# Different artificial feeding strategies shape the diverse gut microbial communities and functions with the potential risk of pathogen transmission to captive Asian small-clawed otters (*Aonyx cinereus*)

Yuanda Gao,[1] Hangyu Zhang,[1] Dapeng Zhu,[2] Long Guo[1]

**ABSTRACT** Captive otters raised in zoos are fed different artificial diets, which may shape gut microbiota. The objective is to evaluate the impacts of two different artificial diets on microbial communities and function capabilities and short-chain fatty acid (SCFA) profiles in healthy otters' feces. A total of 16 Asian small-clawed otters in two groups ($n = 8$) were selected. Group A otters were fed raw loaches supplemented with commercial cat food (LSCF) diet, and group B otters were fed raw crucian diet. The communities and functional capabilities of microbiota in feces were assessed with metagenomic sequencing. Captive otters fed two kinds of diets possessed different gut microbial communities and functional capabilities. Various pathogenic bacteria, like *Escherichia coli* and *Clostridium perfringens*, were enriched in the samples from the two groups, respectively. Most of the differential pathways of nutrient metabolism were significantly enriched in group A, and the distributions of carbohydrate enzymes in the two groups significantly differed from each other. Multiple resistance genes markedly accumulated in fecal samples of the group A otters with LSCF diet. Higher concentrations of SCFAs were also observed in group A otters. Two feeding strategies were both likely to facilitate the colonization and expansion of various pathogenic bacteria and the accumulation of resistance genes in the intestines of captive otters, suggesting that risk of pathogen transmission existed in the current feeding process. Commercial cat food could supplement various nutrients and provide a substrate for the production of SCFAs, which might be beneficial for the otters' intestinal fermentation and metabolism.

**IMPORTANCE** Captive otters fed with different diets possessed distinct gut microbial communities and functions, with the enrichment of several pathogens and multiple resistance genes in their gut microbiota. The current artificial feeding strategies had the possibility to accelerate the colonization and proliferation of various pathogenic bacteria in the intestines of otters and the spread of resistance genes, increasing the risk of diseases. In addition, supplementation with commercial cat food had benefits for otters' intestinal fermentation and the metabolism of gut microbiota.

**KEYWORDS** Asian small-clawed otter, diet, gut microbiota, metagenomic sequencing, short-chain fatty acids

O tters belong to the Mustelidae family of the Carnivora order, a semi-aquatic carnivore and a top predator living in freshwater aquatic ecosystems, playing a vital role in maintaining ecosystem functions (1, 2). There are 13 species and 7 genera of otter in the world (3). The Asian small-clawed otter is a species of the genus *Aonyx*, a second-class protected animal in China, distributed in several southern provinces of

**Peer Reviewers** Shengru Wu, Northwest Agriculture & Forestry University, Yangling, China; Gang He, Northwest University, Xi'an, Shaanxi, China

Address correspondence to Long Guo, guolong@lzu.edu.cn, or Dapeng Zhu, zhudapengsuper@163.com.

The authors declare no conflict of interest.

See the funding table on p. 16.

China, and inhabits rivers, streams and shallow waters along estuaries in the wild (4). The sources and types of the diets of wild otters are diverse and variable. Fish is the main prey category in the diet of wild otters, which also prey on clams, crabs, crayfish, amphibians, insects, and various small mammals when fish availability decreases (5–7).

However, except for the otters distributed in the wild, there are also a certain number of otters kept in captivity at present for ex situ conservation, offering a chance to understand their living habits and characteristics. The diet of captive otters is relatively single and affected by feeding management and also might not be suitable and beneficial for the otter's health. Previous studies on various captive animals have revealed that diet ratio was one of the main causes of several diseases occurring in captive animals. The diet of polar bears in zoos had a higher protein content compared to the diet in the wild, which may be a contributing factor to the end-stage renal disease and cardiovascular disease occurring in captive polar bears (8, 9). The investigation on Vancouver Island marmots (*Marmota vancouverensis*) indicated that artificial diet-induced captive marmots have higher adipose tissue reserves, leading in turn to cardiovascular disease, the primary cause of mortality in captive marmots (10).

At present, otters in various captive conditions or zoos are often given different diets, among which the nutrition supply may vary greatly. Substantial prior studies have documented that diet is an important external factor deeply affecting the makeup and diversity of gut microbiota (11, 12). As a result, the dietary difference of captive otters might lead to differential gut microbial flora. Moreover, some captive otters have been supplementarily fed commercial pet food which contains starch, dietary fiber, vitamins, minerals, and other plentiful nutrient substances. We lack a clear understanding of the impact of feeding dry commercial pet food on the health and gut microbiota of otters. A previous study on canines indicated that commercial dry food and meat-based diets seem to induce different fecal microbiota compositions and that commercial food can reverse changes in fecal microbiota caused by short-term meat-based diets (13). Consequently, a variation of the microbial community might have occurred in the intestine of these captive otters fed pet food.

Gut microbiota is composed of a diverse range of symbiotic gut bacteria and massive other gut microorganisms (14). Apart from playing a crucially important role in nutrient digestion and metabolism (15, 16), it has been proven to be linked to the onset and development of numerous diseases and involved in the modulation of host health, depending on multiple microbiota-derived metabolites (17, 18), such as short-chain fatty acids (SCFAs) (19). The alteration of gut microbiota can be associated with some diseases in humans (20, 21), including obesity (22), type 2 diabetes (23), and inflammatory bowel disease (24). Therefore, we believe that the various artificial diets could have different potential effects on the health and growth of captive otters by influencing their gut microbiota.

To the best of our knowledge, there remains a scarcity of research on captive Asian small-clawed otters, including their diet and gut microbiota, although a few studies have also been carried out using other otter species (25, 26). Accordingly, in this work, we selected two groups of captive Asian small-clawed otters with distinct dietary patterns and collected their feces. Metagenomic sequencing was performed to characterize and compare the structures and functions of gut microbial communities in two groups. This trial was undertaken to identify the differences and variability of gut microbial ecosystems in captive otters with different feeding strategies and nutrient supply, and then to explore the potential effect of artificial diet on otters' health. This could be essential and directive for improving otters' feeding management and ex situ conservation strategies in the future.

**TABLE 1**  Nutrient contents of crucian, loach, and cat food in this research

| Item | Crucian | Loach | Cat food |
|---|---|---|---|
| Nutrient contents (% of DM[a]) | | | |
| DM | 27.25 | 27.34 | 91.78 |
| Ash | 11.90 | 9.84 | 9.21 |
| Ether extract | 16.63 | 24.39 | 18.26 |
| Crude protein | 55.46 | 59.16 | 19.01 |
| Gross energy (MJ/kg of DM) | 27.13 | 31.48 | 25.43 |

[a]DM, dry matter.

## MATERIALS AND METHODS

### Animals, diets, and experimental design

Sixteen healthy Asian small-clawed otters (*Aonyx cinereus*) with an average age of 6 years were selected in this study. Otters from two groups were given two completely different kinds of foods, respectively: group A otters were fed mainly raw loaches (*Misgurnus anguillicaudatus*) supplemented with commercial cat food (LSCF diet; loaches: 84%, commercial cat food: 16%), 600 g/day; group B otters were fed raw crucians (RCs) (*Carassius auratus*) (RC diet), 500 g/day. Otters in the two groups were raised in similar environments and were given the same management.

### Feed sampling and analysis

The feed samples of two groups were randomly collected from the normal diet of otters and then stored at −20°C until nutrient contents analysis. Crucian and loach samples were freeze-dried at −60°C for 76 h, and cat food samples were first dried in a forced-air oven at 65°C for 48 h. Dry matter (DM), crude protein (CP), ether extract (EE), also called crude fat (CF), and ash content of feed samples were measured as described previously (27). The gross energy (GE) content of the dried samples was analyzed by combustion in an adiabatic bomb calorimeter (IKA C3000, Germany). The nutrient compositions and contents of diets are detected and presented in Table 1, and the otters' intakes of diet and several nutrients in two groups are shown in Table 2.

### Fecal sample collection

Fresh fecal samples of otters were collected and stored with sterilized fecal collection tubes without contamination and then transported to the laboratory immediately with dry ice. These samples were stored at −80°C until DNA extraction. All samples were collected from April to June 2022.

**TABLE 2**  Diet and nutrient intake of otters fed LSCF[a] diet in group A or RC diet in group B[a]

| Item | Group | |
|---|---|---|
| | A (LSCF diet) | B (RC diet) |
| Diet intake (g/day) | | |
| Crucian | | 500 |
| Loach (84%) | 504 | |
| Cat food (16%) | 96 | |
| Nutrient intake (g/day) | | |
| Dry matter | 225.90 | 136.25 |
| Ether extract | 49.70 | 22.66 |
| Crude protein | 98.27 | 75.56 |
| Gross energy (MJ/day) | 6.58 | 3.70 |

[a]LSCF, raw loaches supplemented with commercial cat food; RC, raw crucian.

## DNA extraction, library construction, and metagenomic sequencing

All fecal samples were thawed at 4°C before DNA extraction; 200 mg per fecal sample was used for microbial DNA extraction using the E.Z.N.A. Stool DNA Kit (Omega Biotek, Norcross, GA, USA) according to the manufacturer's instructions according to the manufacturer's protocol. The concentration, integrity, and purity of DNA samples were determined using a NanoDrop 2000 spectrophotometer and 1% agarose gel electrophoresis. DNA extract was fragmented to an average size of about 400 bp using Covaris M220 (Gene Company Limited, China) for paired-end library construction. Paired-end library was constructed using NEXTflex Rapid DNA-Seq (Bioo Scientific, Austin, TX, USA). Adapters containing the full complement of sequencing primer hybridization sites were ligated to the blunt end of fragments. Paired-end reads of metagenomic libraries were sequenced on an Illumina NovaSeq 6000 (Illumina Inc., San Diego, CA, USA) at Wefind-bio Technology Co., Ltd. (Wuhan, China) using NovaSeq Reagent Kits according to the manufacturer's instructions.

## Sequence quality control, gene prediction, and genome assembly

The quality control of raw data was performed to trim sequencing adapters, filtering out low-quality reads (reads with N bases, quality scores <20) and short reads (<50 bp) by using fastp (v.0.20.0, https://github.com/OpenGene/fastp) (28), and BWA (v.0.7.9a, http://bio-bwa.sourceforge.net) was used to filter out the reads mapped to the host originated genes to obtain high-quality reads (29). A total of 164-Gb high-quality and clean reads of each sample obtained after quality control were assembled into contigs processed by Megahit (v.1.1.2; parameters: kmer_min = 47, kmer_max = 97, step = 10; https://github.com/voutcn/megahit) (30), which makes use of succinct de Bruijn graphs.

Contigs with a length exceeding 800 bp were obtained finally and used for subsequent analysis. Afterward, MetaGene (http://metagene.cb.k.u-tokyo.ac.jp/) (31) was applied to predict open reading frames (ORFs) of these contigs. The predicted ORFs with lengths being or over 100 bp were retrieved and translated into amino acid sequences using the National Center for Biotechnology Information (NCBI) translation table. All the predicted ORFs were combined to generate the non-redundant microbial gene catalog using CD-HIT (v.4.6.1, http://www.bioinformatics.org/cd-hit/) (32), with clustering criteria of ≥95% identity and ≥90% overlap. Then, all clean reads of each sample were aligned to the non-redundant gene catalog using SOAPaligner (v.2.21, http://soap.genomics.org.cn/) (33) with a criterion of ≥95% identity to get the specific gene abundance information in samples. We finally obtained 2,940,768 non-redundant genes with a 489.4-bp average length and a 579-bp N50 length.

Genome reconstruction or genome binning of gut microbes with metagenomic sequences was carried out by Vamb (v.3.0.5, https://github.com/RasmussenLab/vamb/) (34) for assembling contigs from each sample into metagenomic bins. dRep (v.3.0.0, https://drep.readthedocs.io/) (35) was used to filter the replications of all bins. The completeness and contamination of all bins were estimated by CheckM (v.1.1.3, https://github.com/Ecogenomics/CheckM/) (36) according to the quality evaluation criteria (more than 50% completeness and 10% contamination), and a total of 160 non-redundant bins were identified as metagenome-assembled genomes (MAGs) for downstream analysis. All MAGs were annotated with a taxonomy using GTDB-Tk (v.1.7.0) (37) based on the Genome Taxonomy Database (https://gtdb.ecogenomic.org/).

## Taxonomy and function annotation of genes

To gain insight into the taxonomy profiles of the metagenome of gut microbiota, all the representative sequences of non-redundant genes were aligned to the sequences belonging to bacteria, fungi, archaea, and viruses in the NR database (v.2021.11) using blastp with an $e$ value cutoff of 1e-5 using DIAMOND (v.0.8.35, http://www.diamondsearch.org/) (38). The genes that could not be classified into any taxa were defined as unknown taxa. The Kyoto Encyclopedia of Genes and Genomes (KEGG) and

Cluster of Orthologous Groups of proteins annotation for the representative sequences were performed, respectively, by aligning genes to the KEGG database (v.94.2, https://www.genome.jp/kegg/) (39) with an *e* value cutoff of 1e-5 and eggNOG 4.5.1 database (40) using DIAMOND (v.0.8.35) (38). Carbohydrate-active enzymes (CAZymes) were annotated by aligning genes to the CAZyme database (http://www.cazy.org/) (41) using hmmscan (http://hmmer.org/) with an *e* value cutoff of 1e-5. Antibiotic resistance genes (ARGs) were annotated against the Antibiotic Resistance Genes Database (v.1.1, http://ardb.cbcb.umd.edu/) and the Comprehensive Antibiotic Research Database (CARD) (v.3.0.9, https://card.mcmaster.ca/home) (42) also with DIAMOND (v.0.8.35) (38) through aligning unigenes as abovementioned with an *e* value of ≤1e-30. Differential functional categories and resistance genes were identified by statistical analysis of metagenomic profiles (STAMP) (v.2.1.3) (43).

## SCFA analysis of feces

All samples were thawed at 4℃, then diluted and mixed with distilled water (1-g stool sample diluted with 1-mL distilled water) for 4 h soaking at 4℃, and then centrifuged at $2,000 \times g$ for 10 min to separate the solid residues and liquid. The supernatant fluid was mixed with 25% (wt/vol) metaphosphoric acid (5-mL supernatant fluid and 1-mL metaphosphoric acid) and stored at −20℃. Concentrations of SCFAs (acetate, propionate, isobutyrate, butyrate, and isovalerate) were qualified by gas chromatography (Thermo Fisher Scientific, USA) at the College of Pastoral Agriculture Science and Technology, Lanzhou University.

## Co-occurrence network analysis

The correlation network within group A and B otters was calculated separately by Spearman's correlation coefficient with the R package Spaa (v.0.2.2). To reduce network complexity, those species that had a total abundance of >1 in all samples and presented in at least all eight samples were used for co-occurrence network analysis. For the analysis of MAGs, all MAGs obtained were used for co-occurrence network construction. Spearman's correlation coefficient between two MAGs was considered robust if the absolute *r* value was >0.6 with a corresponding "fdr" adjusted *P* value of <0.05 and the significant and robust correlations between species were defined as an absolute *r* value of >0.8 with a corresponding fdr adjusted *P* value of <0.05. Co-occurrence modules were analyzed by R package igraph (v.1.4.1) with nine modules presented in each group. These correlations obtained above were graphed using Gephi (v.0.10.1) (44) with the layout algorithm of Fruchterman Reingold.

## Statistical analysis

R (v.4.2.2) was used for statistical analysis and visualization of all processed data unless otherwise stated. For clustering heatmaps, the data were normalized using *z*-scores of the abundance of the top 20 bacteria taxa and were visualized by the Pheatmap package in R language. Linear discriminant analysis effect size (LEfSe) analysis of differential bacteria taxa was performed using the microeco R package (v.1.3.0) for data processing, analysis, and plotting.

## RESULTS

## Major nutrient contents of three food sources and two diets

The major nutrient contents including DM, CP, EE, also called CF, ash content, and GE were measured to identify and compare the nutrition values of crucian, loach, and cat food. According to the results, the loach has the highest EE, CP, and GE contents, and the crucian has the highest ash content, while the contents of CP and DM in the loach and crucian are more similar compared to the cat food. The DM content of 91.78% in cat food is the highest clearly because it is a commercial dry pet food with little moisture.

Then, we calculated the daily intake of various nutrients and gross energy for two groups of otters under two different diets based on the nutritional composition of three food sources as shown in Table 2. We could see that the daily intakes of various nutrients (DM, EE, CP, and GE) by group A otters are all higher than those of group B otters.

## Overview of samples collected, metagenomic sequencing data, and gene catalogs

The metagenomic sequencing data from 16 Asian small-clawed otter fecal samples were sequenced in this study, including eight otters fed raw LSCF diet in group A and the other eight otters fed RC diet in group B. Metagenomic sequencing of DNA samples generated high-quality clean data after removing low-quality reads and host reads. After *de novo* assembly, gene prediction and filtering of incomplete genes, 2,940,768 complete unique genes were identified. A total of 13,354 core genes co-existed in all samples, and sample B_2 had the highest number of genes with 1,494,638 unique genes (Fig. S1A). Furthermore, Spearman correlation analysis of gene abundance patterns between groups A and B was carried out, indicating that higher similarity of gene abundance patterns was among the same group samples, and the correlations of samples from different groups were lower (Fig. S1B).

## Taxonomic characteristics of gut microbial communities

To investigate the effects of two diets on the gut microbial taxonomic characteristics of otters, the distribution, composition, and fluctuations of microbiota in feces samples obtained from 16 otters were identified by metagenomic sequencing. Matching the metagenomic genes to the Non-Redundant Protein Sequence Database of NCBI for taxonomic annotation, we found that in more than 99.05% and 73.22% of the classified genes assigned to bacteria in groups A and B, respectively, only 1.1% and 26.78% of the remaining genes belonged to Eukaryota, Archaea, and viruses in the two groups (Fig. S1C). Due to the fact that bacteria make up the vast majority of the gut microbiota, we will primarily analyze the bacteria in the intestines of otters. Although almost a quarter of the sequences in group B otters have been annotated to the eukaryotic superkingdom, the vast majority of these sequences are believed to originate from species in the animal's kingdom, which may be related to otters consuming animal-derived food, while only a very small portion of the fungi kingdom species may be of concern to us (detailed data are shown in Table S1). Therefore, we will mainly focus on the bacterial community in the intestines of otters in the present study.

The relative abundance of the different bacteria at all phylogenetic levels was identified among the two groups of otters fed with different diets, and the top 20 bacterial taxa of genus and species in abundance are depicted in Fig. S2A and B; Fig. 1A and B. A total of 438 core genera were found distributed in all 16 samples (Fig. S2F). The most major taxa identified at the genus level in the feces samples from group A were *Clostridium* (32.52% ± 10.87%) followed by *Aeromonas* (22.99% ± 10.69%), while *Clostridium* (61.56% ± 24.66%) and *Romboutsia* (8.64% ± 4.29%) were the two most prevalent genera in group B (Fig. S2A and B). At the species level, we identified different numbers of bacteria species in the fecal samples from the two groups and also found 472 core species shared in all samples with Venn diagram analysis (Fig. 1F). Furthermore, the bacterial communities in the gut of otters from two groups were dominated by several main species such as *Clostridium perfringens* (47.69% ± 16.74% for group A and 69.12% ± 25.12% for group B), *Escherichia coli* (13.38% ±13.79% for A; 0.18% ± 0.24% for B) and *Cetobacterium* somerae (3.56% ± 3.82% for A; 0.22% ± 0.39% for B) (Fig. 1A and B). Although these species were all believed to be pathogenic bacteria associated with a variety of diseases, the above results in this part indicated that the two group's otters possessed distinct relative abundances and compositions of main gut bacteria.

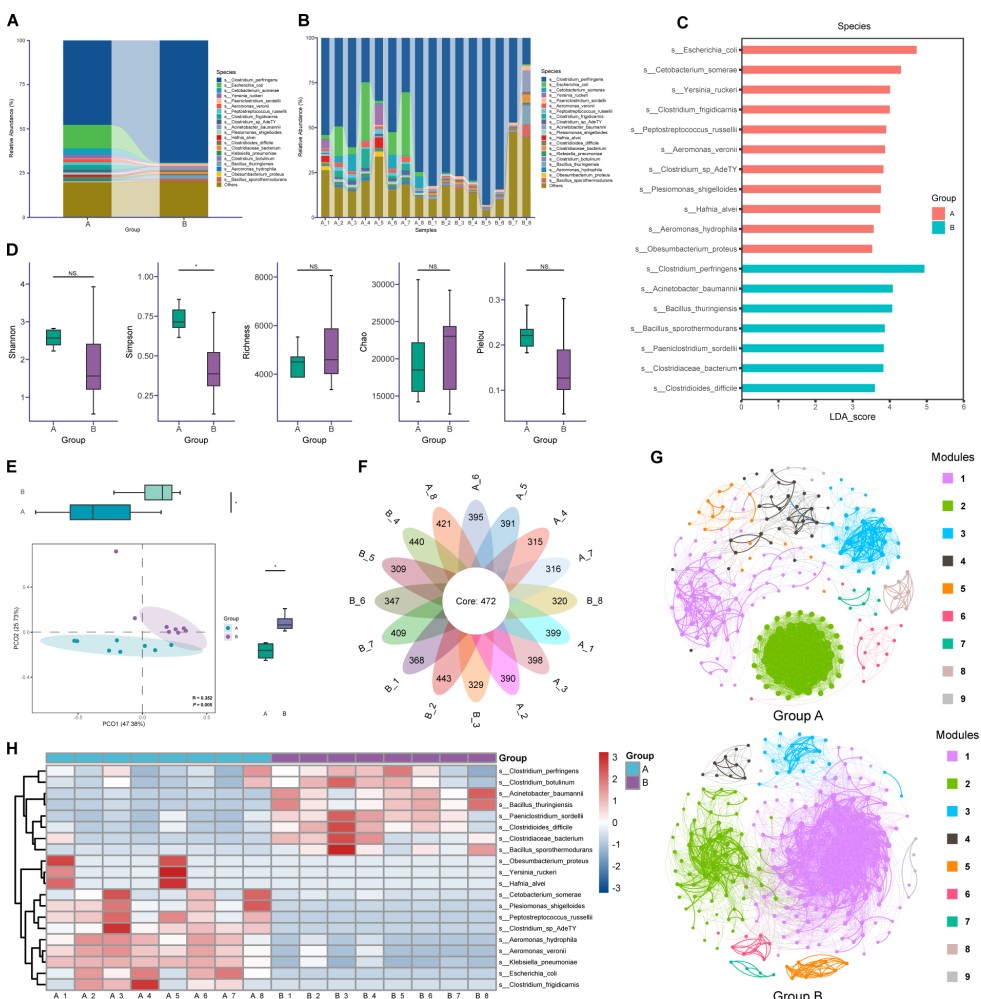

**FIG 1** The compositions and structures of fecal microbial communities of otters from two groups at species level. (A) The top 20 gut bacteria average relative abundances at species level in groups A and B. (B) The top 20 gut bacteria relative abundances at species level in each sample. (C) The linear discriminant analysis (LDA) score (log 10) of differential bacteria at species level (LDA score >3.5 and Kruskal−Wallis test $P$ value < 0.001). (D) The α diversity analysis at species level including Shannon index, Simpson index, richness index, Chao1 index, and Pielou index. (E) Principal coordinate analysis plot of bacteria community at species level based on Bray−Curtis distance ($n = 8$). *$P < 0.05$, by independent-samples $t$-test. (F) Flower plot of Venn analysis of bacteria species number in 16 samples. (G) Co-occurrence network analysis of bacteria community at genus level in groups A and B where nodes are colored according to modules (1–9). Each node represents one bacteria genus; each edge represents a strong and significant positive or negative correlation selected based on the threshold of a Spearman rank correlation coefficient of $P$.adjust <0.05 and $| r | ≥0.8$ between two nodes. The size of each node is proportional to the degree of the bacteria taxa. The thickness of edges is proportional to the value of the Spearman correlation coefficient. (H) Clustering heatmap showing the relative abundances of the top 20 bacteria species taxa in each sample standardized using $z$-score method. NS, no significance.

## Discrepancies in gut microbial communities between the two groups

Significantly higher values of alpha index were observed in group A compared to group B in the estimations of α diversity at genus and species levels (Fig. S2D; Fig. 1D). A higher Simpson index was observed in group A at the two taxonomy levels, and a higher richness index was observed in group B at the genus level. Principal coordinate analysis (PCoA) based on Bray−Curtis with an analysis of similarities was performed to characterize the divergence of the gut microbial community of two groups at genus and species level as well (Fig. S2E; Fig. 1E). PCoA score plots for the two groups showed a clear separation that bacterial communities from one group clustered together and separated

from those in the other group, suggesting that the inter-group differences between two groups were significantly greater than the intra-group differences ($r = 0.352$, $P = 0.005$, at species level) and that the two groups of otters harbored distinct bacterial floras. Clustering analysis was used to further determine the similarities and differences in the top 20 microbial species abundances between samples (Fig. 1H).

To further explore the differential gut bacteria between the two groups, we performed LEfSe to screen significantly different biomarkers discriminating the two groups. Using more stringent criteria (LDA score of >3.5 and Kruskal-Wallis test $P$ value of < 0.001), 18 bacterial genera and species were identified as significantly differentially represented between the two groups respectively (Fig. S2C; Fig. 1C). At the species level, 11 bacterial species were significantly enriched in group A, and group B was distinguished from the other 7 bacteria (Fig. 1C). For instance, *Escherichia coli* was notably enriched in group A but depleted in group B, whereas *Clostridium perfringens* was more prevalent in group B but declined in group A.

We constructed a co-occurrence network analysis at the species level to characterize the interactions of gut microbiota in each group. The correlation networks were analyzed by Spearman's correlation coefficient, and the significant correlations were visualized in Fig. 1G. These species of gut microbiota in the two groups were clustered into nine modules, respectively. The correlation network of group B (nodes: 260, edges: 4,831) possessed more nodes and edges than group A (nodes: 229, edges: 2,302) at the species level, filtered in stricter threshold ($P < 0.05$, $|r| > 0.80$). The correlations between nodes and modules in the group A species co-occurrence network are less tight than those in group B, especially module 2 in group A nearly completely separated from other modules without any strong correlation. Overall, the result of the co-occurrence network analysis demonstrated that the connection of gut bacteria in group B was closer.

## Reconstruction of microbial metagenomic genomes

Metagenomic contigs from 16 fecal samples were reassembled and clustered into 160 microbial genomes belonging to strains with a threshold of >50% completeness and a contamination of ≤10% (Fig. 2I). Then we used the Nucleotide Sequence Database of NCBI for taxa annotation of these reconstructed genomes; 160 MAGs were classified into Bacteria and Archaea superkingdom. Additionally, 147 of these MAGs were annotated to eight different phyla, 113 MAGs were aligned to 34 genera, but only 60 MAGs were annotated to species level (Fig. 2A through D; Table S2). We presented the compositions and relative abundances of the top 20 MAGs belonging to four taxa within each sample as shown in Fig. 2E. Meantime, the analysis of the top 20 MAGs' average relative abundances in the two groups is shown in Fig. 2F, indicating the distribution patterns of these MAGs in two groups were significantly different (Fig. 2G). A co-occurrence network analysis of MAGs in two groups was also performed, showing a result that the MAGs in group B connected more tightly and were more stable, similar to the result of previous analysis at the species taxonomic level (Fig. 2H). STAMP analysis revealed several MAGs possessed different abundances in the two groups. Five MAGs like "S15C2549" and "S14C2549" were differentially enriched in group A. All of the five MAGs belonged to *c_Gammaproteobacteria*, which holds more than 20 genera containing some taxa that infect humans and animals (45), such as the bacterium *Escherichia coli*, well-known pathogens *Salmonella*, *Yersinia*, *Vibrio*, and *Pseudomonas* (46). Meanwhile, there were also other five MAGs markedly existing in group B. Four of them were annotated to *f-Clostridiaceae*, and the other MAG, "S4C5569," was aligned to *s_Clostridium perfringens* (Fig. 2J).

## Functional characteristics and differences of gut microbial communities in otters from the two groups

To characterize the functions of the gut microbiota of these captive otters, metagenomic genes were aligned to the KEGG, eggNOG, and CAZymes databases. The KEGG

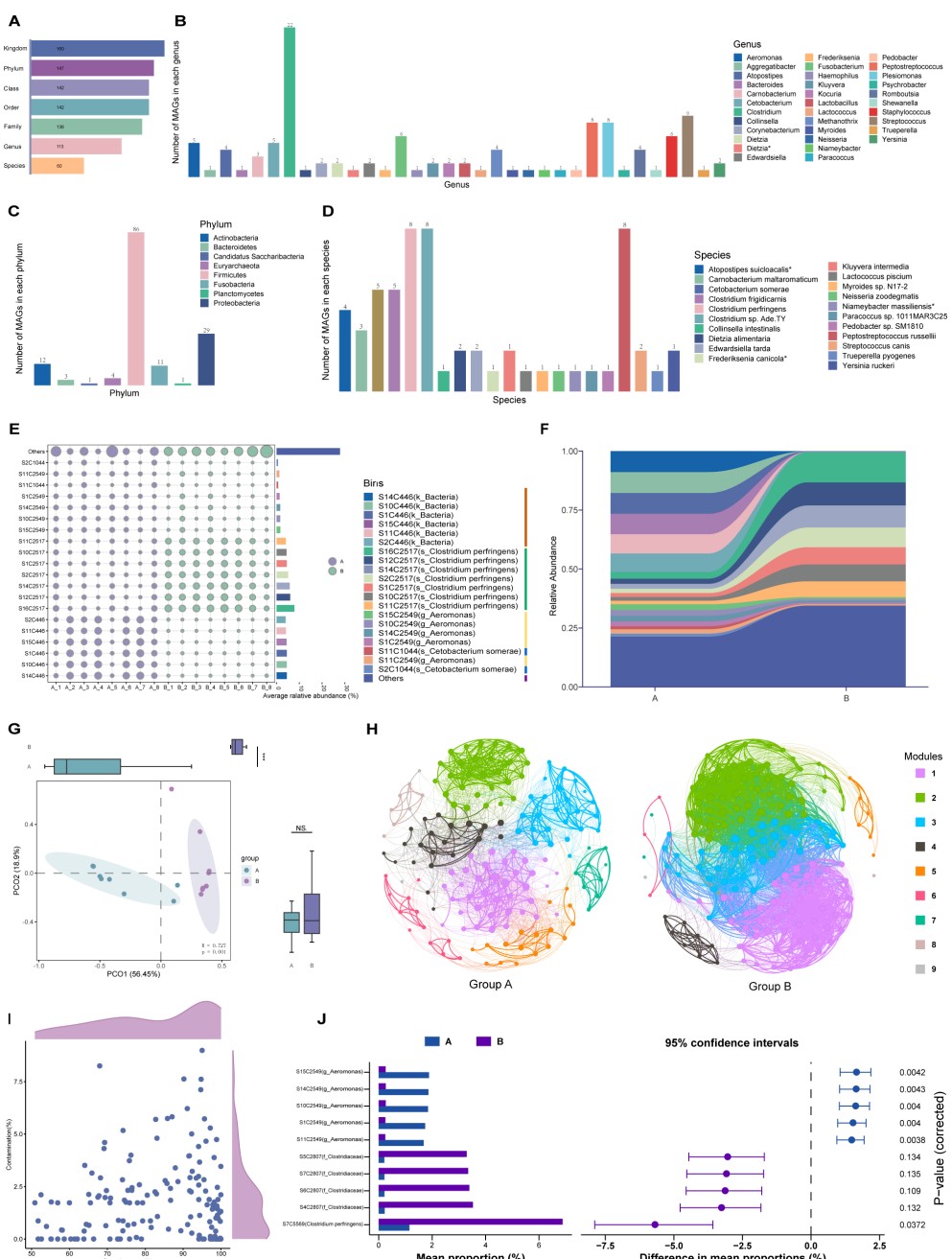

FIG 2 The classifications and compositions of MAGs in two groups and the analysis of differential MAGs. (A) Summarization of the number of MAGs annotated into different taxonomic levels. (B–D) The number of MAGs annotated in each phylum, genus, and species taxon. The MAGs that could not be annotated are not shown. (E) The relative abundances of top 20 MAGs in different samples. The top 20 abundant MAGs were annotated in four taxa marked by four different colors in the legend. (F) The top 20 MAGs' average relative abundances in groups A and B. (G) Principal coordinate analysis plot of MAGs within two groups based on Bray−Curtis distance ($n = 8$). ***$P < 0.001$ by independent-samples $t$-test. (H) Co-occurrence network analysis of bacteria community at MAG level where nodes are colored according to modules (1–9). Each node represents one MAG; each edge represents a strong and significant positive or negative correlation selected based on the threshold of a Spearman rank correlation coefficient of $P$.adjust <0.05 and | $r$ | ≥0.6 between two nodes. The size of each node is proportional to the degree of the MAGs. The thickness of edges is proportional to the value of the Spearman correlation coefficient. (I) Distribution of completeness and contamination of all obtained MAGs. (J) Extended error bar plot of STAMP analysis presenting the significantly differential MAGs in the two groups.

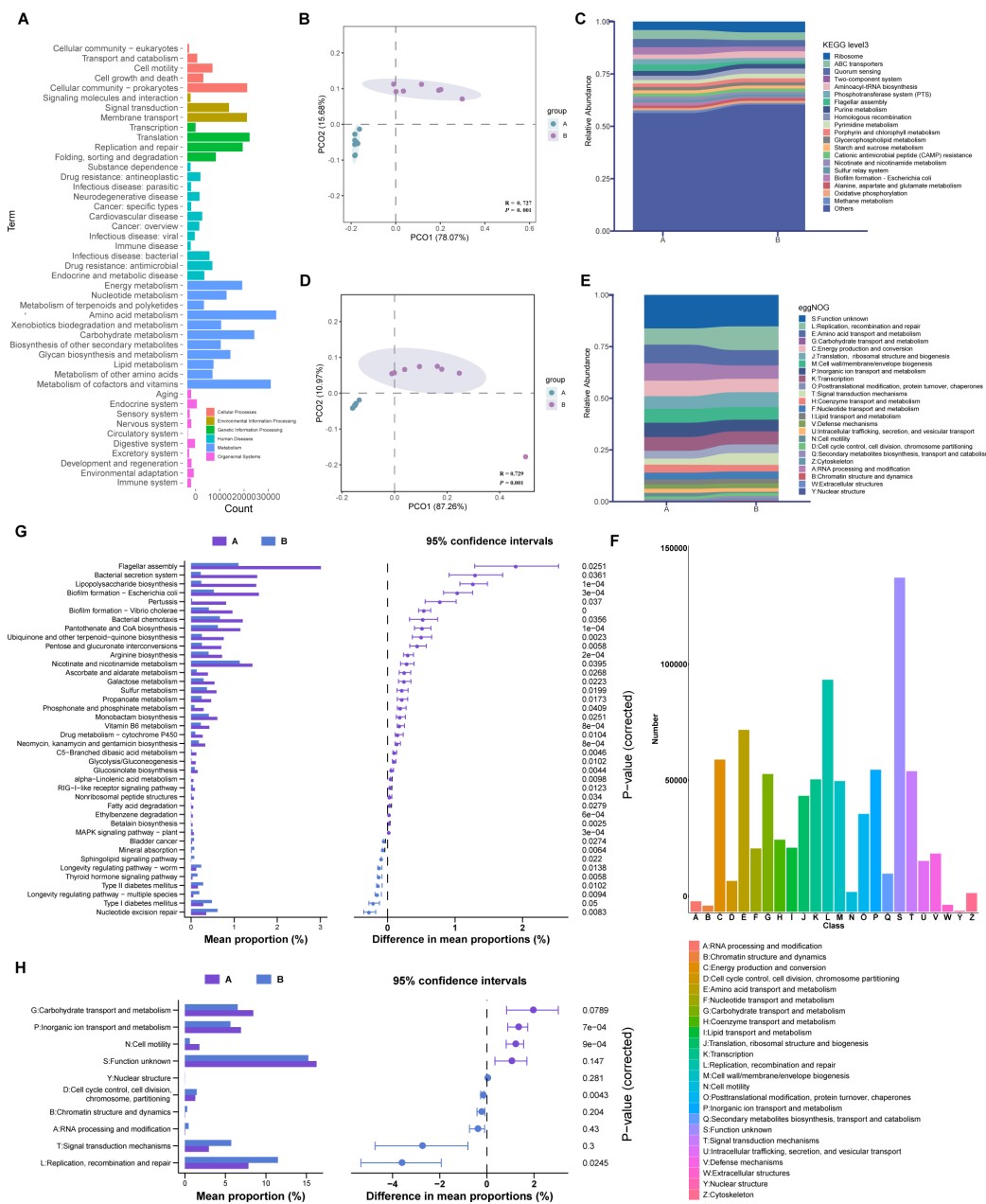

**FIG 3** The abundance and the differential analysis of KEGG and eggNOG annotations. (A) KEGG classifications for the fecal metagenome. (B) Principal coordinate analysis plot of KEGG pathways based on Bray−Curtis distance (*n* = 8). (C) The compositions and abundances of top 20 KEGG pathways at the third level in the two groups. (D) Principal coordinate analysis plot of eggNOG categories based on Bray−Curtis distance (*n* = 8). (E) The compositions and abundances of the top 20 eggNOG functional categories in the two groups. (F) eggNOG classifications for the fecal metagenome. (G) Extended error bar plot of STAMP analysis presenting differential KEGG pathways in the two groups. (H) Extended error bar plot of STAMP analysis presenting differential eggNOG functional categories in the two groups.

enrichment analysis of metagenomic genes from all samples confirmed 438 third-level pathways, in which 388 pathways were shared in the two groups. All of these pathways obtained belonged to 6 first-level categories and 45 second-level categories (Fig. 3A). "Metabolism" was found to be the most predominant first-level pathway in the two groups (Fig. 3A), and "amino acid metabolism" was the most dominant second-level category (Fig. 3A). At the third level, the relative abundances of 20 top pathways were visualized (Fig. 3C). Meanwhile, PCoA of KEGG level 3 presented a noted separation (Fig.

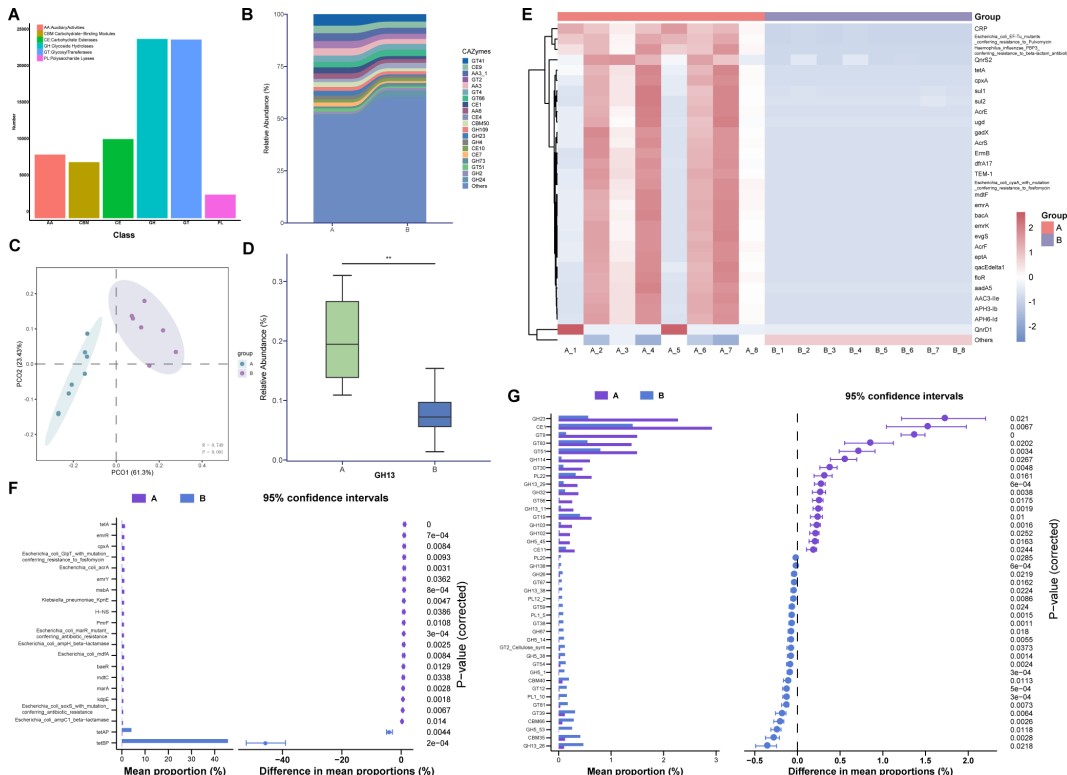

**FIG 4** The abundance and the differential analysis of CAZymes and resistance genes in two groups. (A) Carbohydrate-active enzyme (CAZyme) classifications for the fecal metagenome. (B) Compositions and relative abundances of top 20 Carbohydrate-active enzymes annotated by CAZy database. (C) Principal coordinate analysis plot of CAZyme families based on Bray−Curtis distance ($n = 8$). (D)The relative abundance of GH13 CAZymes in the two groups. **$P < 0.01$ by Mann–Whitney $U$ test. (E) Clustering heatmap representing the relative abundances of the top 30 resistance genes in the two groups standardized using the $z$-score method. (F) Extended error bar plot of STAMP presenting differential CAZymes in the two groups (G) Extended error bar plot of STAMP presenting differential resistance genes in the two groups.

3B). In addition, eggNOG functional annotation identified 24 categories in level 1 with "replication, recombination, and repair" being the most abundant in all categories except "function unknown" (Fig. 3F). Functions connected with the transport and metabolism of nutrients were also highly abundant, such as "amino acid transport and metabolism" and "lipid transport and metabolism." Similarly, the separation of distribution and the differences in relative abundances of eggNOG categories of the two groups were verified as well (Fig. 3D and E).

Subsequently, STAMP was used for the identification of functions with significant differences between groups. The significantly differential KEGG pathways are shown in Fig. 3G. There were 31 KEGG functional pathways with marked enrichment in group A and 9 pathways enriched in group B. For instance, "lipopolysaccharide biosynthesis" correlated with the production of Gram-negative bacteria. Moreover, some pathways related to the production of opportunistic pathogens and diseases, like "biofilm formation–*Escherichia coli*," were all enriched in group A, whereas pathways like "thyroid hormone signaling pathway" and "nucleotide excision repair" were enriched in group B (Fig. 3G). Additionally, we also discovered that functions involving "inorganic ion transport and metabolism" and "amino acid transport and metabolism" were remarkably abundant in group A (Fig. 3H).

For CAZyme profiles, 79,446 CAZyme families were identified as belonging to six function categories with the annotation of the CAZy database, in which glycoside hydrolase (GH) and glycosyltransferase were the two most annotated categories (Fig. 4A). The relative abundances of the top 20 CAZyme families are presented in Fig. 4B. The PCoA exhibited a considerable difference between the two groups (Fig. 4C). Additionally,

STAMP showed that GH23 and CE1 were the most significantly enriched CAZymes in group A (Fig. 4G). Furthermore, the total relative abundance of GH13 CAZyme in the two groups was focused on its specific function in starch degradation, showing that the relative abundance of GH13 was significantly higher in group A (Fig. 4D).

Furthermore, metagenomic genes were also classified to the CARD to annotate the ARGs that existed in otters' gut microbiota. We identified 232 different antibiotic resistance ontologies (AROs) in all samples, with significantly different abundances in all samples and most AROs significantly enriched in group A (Fig. 4E and F). The most abundant 10 AROs conferred resistance to eight groups of antibiotics comprising macrolides, elfamycins, aminoglycosides, β-lactams, tetracyclines, quinolones, sulfonamides, and polypeptides (Table S3).

## The associations between SCFA profiles and differential gut bacteria taxa

According to the existing reports, the type and nutrient contents of diet have been shown to exert an influence on the profiles of SCFAs in feces (47). Thus, we quantified the concentrations of different SCFAs (including acetate, propionate, butyrate, and valerate) in feces of otters using gas chromatography to seek the differences in concentrations of SCFAs among the groups. Comparing the total abundance of SCFAs in otters' feces of the two groups, we found that the SCFA concentrations in group A otters' feces were higher (Fig. 5A), and acetate was the most common SCFAs in all samples, followed by propionate. In addition, the concentrations of acetate, propionate, butyrate, and valerate were all notably more abundant in group A ($P < 0.05$). *Roseburia* and *Butirivibrio* are generally considered to be genera-producing butyric acid. We focused on the concentrations of the two butyrate-producing bacteria genera. The concentrations of *Roseburia*

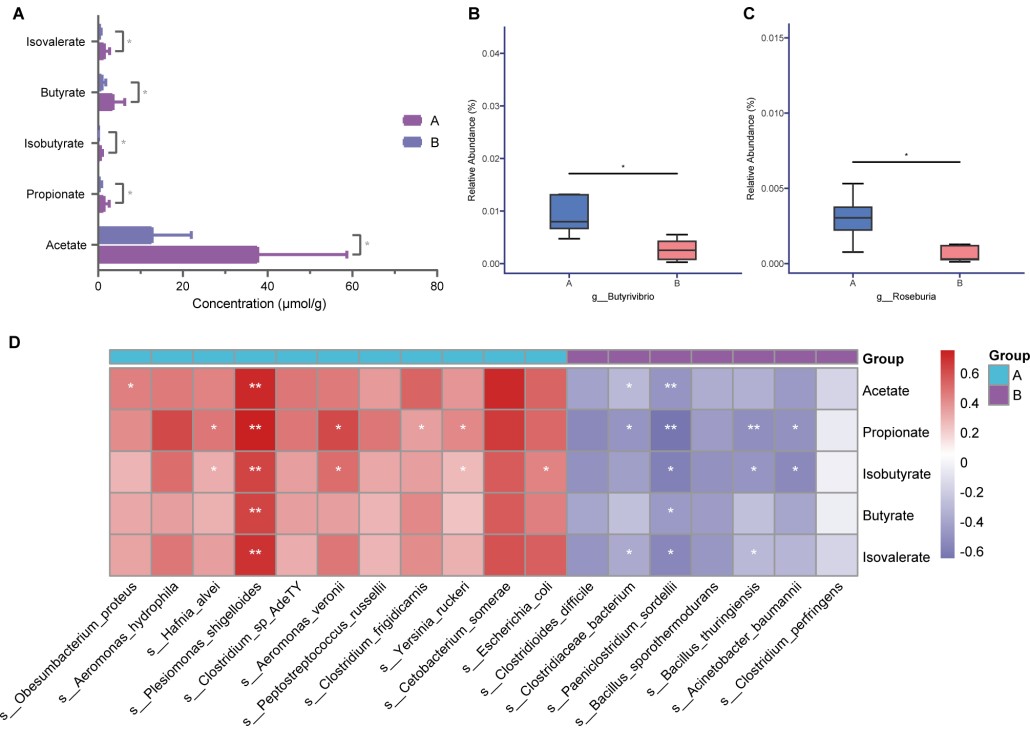

**FIG 5** The VFA concentrations and the correlation analysis between the VFA concentrations and the differential bacteria. (A) The concentrations of acetate, propionate, butyrate, isobutyrate, and isovalerate. *$P < 0.05$ by independent-samples *t*-test followed by Mann–Whitney *U* test. (B) The abundance of g_Butirivibrio in the two groups. *$P < 0.05$ by Mann–Whitney *U* test. (C) The relative abundance of g_Roseburia in the two groups. *$P < 0.05$ by Mann–Whitney *U* test. (D) Spearman correlation analysis between the VFA concentrations and the differential bacteria taxa relative abundances at species level. Red denotes a positive correlation and blue denotes a negative correlation. The intensity of the color is proportional to the strength of the Spearman correlation. *$P < 0.05$, **$P < 0.01$.

and *Butirivibrio* were both significantly higher in group A (Fig. 5B and C). To further investigate whether the differences in fecal SCFA contents were caused by the differential gut microbial species, the Spearman correlation coefficient between SCFAs in feces and the differential gut bacteria species was calculated (Fig. 5D). We found that almost all of these differential bacteria species enriched in group A were positively correlated with the concentration of SCFAs, while bacteria species significantly enriched in group B all have a negative correction with the production of SCFAs. Additionally, there was a strong positive correlation between *Plesiomonas shigelloides* which significantly existed in LSCF diet group samples and the concentrations of all SCFAs, while *Paeniclostridium sordellii* negatively correlated with the concentrations of all SCFAs.

## DISCUSSION

Understanding the influences of diet on the gut microbiota of captive otters and finding out the shortcomings of current artificial feeding strategies and the potential effect of artificial diet on the health of captive otters could contribute to enhancing our knowledge and developing targeted dietary strategies for optimizing the well-being of captive otters. In the present study, a comparison of microbial communities in fecal samples from two groups of Asian small-clawed otters fed the RC diet and raw LSCF diet was performed using metagenomic sequencing. The observations of this study indicated that different artificial feeding strategies have shaped the gut microbiota of captive otters with completely different structures and functions between the two groups and may promote the spread of some pathogenic microorganisms and resistance genes simultaneously, thereby increasing the potential risk of otter infection.

Accumulating evidence has suggested that the structures and the α diversity of gut microbiota could be subject to the diet nutrient contents. For instance, high-sugar, high-fat, and high-protein diets were demonstrated to significantly alter the intestinal microbiota distribution patterns (48). Another study showed that increasing dietary fiber resulted in a shift of microbial community in mice fed the plant-based diet (49). Herein, we compared the gut microbial communities that existed in two groups of otters, observing different microbial compositions and higher Simpson index at genus and species levels in the gut microbiota of group A otters, whereas the richness index was higher in groups at genus level, indicating the microbial community in group B possesses richer bacteria genus taxa, while the evenness of gut microbiota in group A might be higher, considering that the Simpson index can reflect the two dimensions of richness and evenness in a community, especially with a more sensitive response to evenness. This is might because more fat and protein intakes of group A otters fed LSCF diet were observed from Table 2, which provided more available substrates for gut microbes, promoting their proliferation and increasing their diversity and richness. However, higher α diversity was also found in dogs and cats consuming a raw food diet and a natural diet compared with those in the commercial feed group, which is a contradiction with our observations (50–52). This might be caused by the small portion accounted by commercial cat food in otters' diets and the low content of carbohydrate dietary fiber contained in the commercial cat food, not exerting an obvious effect on the diversity of the gut microbial community.

The enrichment of KEGG and eggNOG categories of gut microbiota was correlated with gut microbial richness and the bioavailability of nutrients in the intestine tract. The higher microbial mass in the intestines of group A otters also signifies the probability of the enrichment of microbiota-related metabolic pathways, especially the metabolism pathways related to diet nutrients. This conjecture is in line with our observation that most of the differential functional capacities of the gut microbiota were enriched in group A, which included LSCF diet-feeding otters, and mainly related to the metabolism of nutrients (amino acids, carbohydrates, and vitamins), genetic information processing, and the propagation of some pathogenic bacterial species in the gut. The enrichment of nutrient metabolic pathways in group A may come down to the higher fat, protein, and energy intake under the LSCF diet compared to group B. Moreover, other various

nutrient substances also contained in this diet, such as carbohydrates, vitamins, and minerals, might be partially supplied by commercial cat food.

The profound differences in the enrichment of carbohydrate enzymes between the two groups could be attributed to the introduction of different dietary carbohydrates, which have been reported to drive the alterations of the microbiome and the compositions of associated CAZyme on various time scales (53). GH13 is a principal enzyme family for starch degradation in the α-amylase family (54). A higher abundance of GH13 in group A suggests that these otters may exhibit an enhanced utilizing capability of starch. GH23 family was the differential CAZyme that was most significantly enriched in group A and is considered to have chitinase and lysozyme activities. However, it is unclear what role it plays in gut microbiota of captive otters.

Feeding raw fish or other raw aquatic products would induce gut microbiota dysbiosis (55) and might increase the risk of gastrointestinal diseases (56) and system and chronic diseases (57) by the transmission of numerous pathogenic bacteria from feed to otters. As claimed by LEfSe, several opportunistic pathogenic bacteria species, like *Escherichia coli*, *Aeromonas veronii, and Aeromonas hydrophila,* were significantly enriched in group A by the result of MAG analysis. As we all know, *Escherichia coli* is a well-known opportunistic pathogen that can cause intestinal diseases and extra-intestinal infections in various parts of the body when the host is immunocompromised (58). *Aeromonas veronii* is a highly pathogenic bacteria that widely exists in the environment of humans, animals, and aquatic animals and can cause a variety of diseases (59). The bacteria *Aeromonas hydrophila* is a zoonotic bacterial pathogen that frequently causes disease and mass mortalities among cultured and feral fishes (60). Additionally, two other pathogenic bacteria, *Yersinia ruckeri* (61) and *Plesiomonas shigelloides* (62), were enriched in group A as well. The discovery of the enrichment of these pathogenic bacteria in group A otters was somewhat unexpected. Accumulation of these pathogenic microorganisms indicated otters in this experiment may be facing a health threat and have a high risk of developing a disease. Owing to the fact that most of these enriched bacteria were the pathogens of fish diseases, we speculated that the loaches may harbor certain pathogenic bacteria, thereby spreading the pathogenic microorganisms into the otter's intestine with the foraging behavior. One of the principal causes is that loaches, with intensive loach culture quickly expanding, are more vulnerable to various bacterial infections, most commonly caused by *Aeromonas* spp., which was found markedly enriched in otters fed loaches in this research (63). Furthermore, potential pathogen contamination during the stage of transportation and storage cannot be ignored. The consumption of fresh fish allowed the naturally occurring microorganisms existing in these foods to colonize the gut with a high likelihood (64), promoting infectious bacteria to predominate in the microbiome community of otters consuming this type of food. Interestingly, even though so many bacteria could cause diseases that were abundant in otters' intestines, all of the otter individuals in this experiment were in good health, and the occurrence of any diseases in these otters in this experiment was not observed. However, it is unknown whether the pathogenic bacteria observed in this study will cause intestinal diseases in the future. There is a dearth of knowledge on the specific roles of these bacteria in the gut of Asian small-clawed otters, and it is possible that the fish-related bacteria in these healthy otters may function as part of the normal gut microbiota of otters. Nonetheless, the result that multiple pathogenic bacteria enriched in the intestine of otters could also serve as a reminder that some measures should be taken to detect the level of pathogenic bacteria in the diet fed to otters to avoid these pathogens' spreading and infection as far as possible.

We also detected the presence of several pathogenic microorganisms with significant enrichment in the fecal samples of group B otters fed RC diet, including *Clostridium perfringens* and *Paeniclostridium sordellii. Clostridium perfringens* is an extreme pathogen of humans and livestock, causing wound infections like gas gangrene, enteritis, and enterotoxemia (65). However, *C. perfringens* was found to be present in high abundance in healthy dogs as well as other carnivorous species and is considered a

common commensal in carnivores. *Paeniclostridium sordellii* is a Gram-positive anaerobic bacterium that opportunistically causes acute infectious diseases in humans and animals, involving myonecrosis and enterotoxaemia (66). These pathogenic microorganisms may represent a serious health hazard to the group B otters fed RC diet. Therefore, it is essential and critical to consider the potential risk of bacterial contamination in the RC diet and address it through monitoring food pathogenic microorganism content and utilizing appropriate storage and preparation techniques (51).

Fishes and livestock are often treated with antibiotics during the feeding period to maintain health and productivity in aquaculture or husbandry (67). Antibiotic exposure is thought to contribute to raw fish and meat carrying resistance genes, which could be indirectly transferred to the intestinal tract of animals, contributing to the enrichment of resistance genes (68). In addition, one prior work demonstrated that high-sugar, high-fat, and high-protein diets promoted the amplification and transfer of exogenous ARGs among intestinal microbiota in mice (48). The enrichment analysis of resistance genes in this research showed that the total abundance of resistance genes in group A was significantly higher than that in group B. We speculated that loaches may carry a large number of bacteria with antibiotic resistance genes so that these resistance genes enter the intestine of the otter along with the foraging behavior. At the same time, the higher fat and protein contents of loaches offered chances for bacterial reproduction, resulting in the enrichment of resistance genes in the otters consuming loaches. The accumulation of resistance genes in group A otters indicated an increasing risk of compromising the treatment of infections and higher morbidity. However, one thing we cannot ignore is that although we have considered food or feeding strategy as the main risk factor for the transmission and enrichment of pathogen and resistance genes in the above discussion, there are still other factors that may produce similar results, such as living environment and artificial water bodies, as well as possible human transmission due to the fact that otters were raised in zoos, having the chance of contact with people.

Commensal gut resident microbiota has an essential role in maintaining barrier integrity by producing beneficial metabolites, such as butyrate, one of the most prevalent SCFAs, which has been described as an essential factor for proliferating and maintaining the gut barrier and strongly influences the microbial environment (17). Undigested carbohydrates can provide substrates for the production of SCFAs mainly by the fermentation of bacteria in the colon, whereas the branched-chain fatty acids (BCFAs) are typically generated from protein fermentation (69, 70). Commercial pet foods usually contain large quantities of carbohydrates, typically 46%–74% on a DM basis, and a small proportion (>4% DM) of dietary fiber (71), suggesting more carbohydrate intake of group A otters fed LSCF diet. Therefore, higher concentrations of SCFAs were detected in group A. In addition, higher contents of *Roseburia* and *Butirivibrio* correspond to the concentrations of SCFAs in group A. Higher isobutyrate concentrations observed in group A feces indicated more protein to produce BCFAs as well, which is consistent with the measurement of nutrient intake of otters as Table 2 shows. According to the correlation analysis, significantly differential bacteria species enriched in group A are mostly positively correlated with the concentration of SCFAs. *Cetobacterium somerae*, which belongs to *Fusobacteria*, was found to be enriched in group B and significantly related to all five SCFAs detected in our study. *C. somerae* is the main species constituting the phylum of *Fusobacteria* in fish microbiota and has been identified in the microbiota of many fish species, which can produce a high amount of acetate and only a minor of propionate and butyrate (72).

The enrichment of pathogenic bacteria-related pathways in the intestine of otters is caused by the excessive colonization of pathogenic bacteria in the intestine, consistent with the analysis results of LEfSe. In addition, we revealed that type 1 and type 2 diabetes-related pathways were significantly enriched in group B, suggesting this gut microbiota may induce host to have a greater possibility of developing chronic diseases. High blood or fecal levels of SCFA have been associated with the alleviation or prevention of type 2 diabetes, cardiovascular disease, and obesity. SCFA production by gut

microbiota is considered to be one of the mechanisms through the gut microbiome that plays a role in the prevention or alleviation of symptoms of these diseases (73–75). It has been shown that the consumption of dietary fiber helps to maintain the diversity of gut microbiota and promotes the production of short-chain fatty acids through fermentation, which is generally recognized as healthy, safeguarding intestinal health (76, 77). Taken together, these findings suggest that supplementing cat food to the diet of otters may not have a negative impact on their gut microbiota and hindgut fermentation, and may even improve their diet nutrient composition and carbohydrate intake, thereby regulating the fermentation and metabolism activity of their gut microbiota. This discovery will offer a new insight of making a scientific and fitting diet for captive otters through the perspective of gut microbiota.

## Conclusions

Otters fed two kinds of diets exhibited significantly different gut microbial communities in structures and functions. LSCF diet with higher nutrient contents leads to a richer and more metabolically active microbial flora. The present feed strategies for otters in captivity can accelerate the colonization and proliferation of various pathogenic bacteria in the intestines of otters and the spread of resistance genes, increasing risk of diseases. The higher concentrations of SCFAs detected in group A with LSCF diet are related to the enrichment of SCFA-producing bacteria in the intestine with the supplement of commercial cat food, suggesting that supplementing cat food to otters may have a positive effect on their gastrointestinal fermentation and health. Our study demonstrated the clear role of different artificial diets in the gut microbiota and their potential impacts on the health of captive otters. This finding reminds us to pay attention to the risks of foodborne disease infection and transmission in the otter feeding process and thereby underscores the importance of reforming dietary strategies for the well-being of captive otters. However, the most appropriate dietary strategies and nutritional requirements of captive otters remain unclear, requiring further investigations to clarify.

## ACKNOWLEDGMENTS

This research was supported by the Scientific Research Startup Foundation of Lanzhou University (561119219).

Y.G. was involved in data curation, formal analysis, and writing (original draft). H.Z. was involved in methodology, validation, and investigation. D.Z. was involved in conceptualization, investigation, and methodology. L.G. was involved in funding acquisition, investigation, data curation, formal analysis, and writing (review and editing). All authors have read and agreed to the published version of the manuscript.

The authors declare that the research has no potential conflicts of interest.

## AUTHOR AFFILIATIONS

[1]State Key Laboratory of Herbage Improvement and Grassland Agro-Ecosystems, College of Pastoral Agriculture Science and Technology, Lanzhou University, Lanzhou, China
[2]Foping National Nature Reserve, Hanzhong, China

## AUTHOR ORCIDs

Yuanda Gao http://orcid.org/0009-0004-5523-8718
Long Guo http://orcid.org/0000-0003-0882-1420

## FUNDING

| Funder | Grant(s) | Author(s) |
| --- | --- | --- |
| Scientific research start-up foundation from Lanzhou University | 561119219 | Long Guo |

## AUTHOR CONTRIBUTIONS

Yuanda Gao, Data curation, Formal analysis, Writing – original draft | Hangyu Zhang, Investigation, Methodology, Validation | Dapeng Zhu, Conceptualization, Investigation, Methodology, Resources, Supervision | Long Guo, Data curation, Formal analysis, Funding acquisition, Investigation, Writing – review and editing

## DATA AVAILABILITY

All metagenomic data used to support the results of this study have been uploaded to the National Center for Biotechnology Information Sequence Read Archive database under accession number PRJNA1105391.

## ETHICS APPROVAL

This study was approved by the Animal Care and Use Committee of Lanzhou University.

## ADDITIONAL FILES

The following material is available online.

### Supplemental Material

**Fig. S1 (mSystems00954-24-S0001.tif).** The distributions of non-redundant unique genes in samples of two groups.
**Fig. S2 (mSystems00954-24-S0002.tif).** The compositions and differences of fecal microbial communities of otters from two groups in genus level.
**Legends (mSystems00954-24-S0003.docx).** Supplemental figure and table legends.
**Supplemental Tables (mSystems00954-24-S0004.xlsx).** Table S1 to S3.

### Open Peer Review

**PEER REVIEW HISTORY (review-history.pdf).** An accounting of the reviewer comments and feedback.

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
