## [Reviewer comments · mSystems]

Different artificial feeding strategies shape the diverse gut microbial communities and functions with the potential risk of pathogen transmission to captive Asian small-clawed otters (*Aonyx cinereus*)

Yuanda Gao, Hangyu Zhang, Dapeng Zhu, and Long Guo

Corresponding Author(s): Long Guo, Lanzhou University

Review Timeline:

Submission Date:	July 16, 2024
Editorial Decision:	August 26, 2024
Revision Received:	October 28, 2024
Accepted:	November 4, 2024

Editor: Suzanne Ishaq

Reviewer(s): Disclosure of reviewer identity is with reference to reviewer comments included in decision letter(s). The following individuals involved in review of your submission have agreed to reveal their identity: Shengru Wu (Reviewer #1); Gang He (Reviewer #2)

Transaction Report:

DOI: <https://doi.org/10.1128/mSystems.00954-24>

Re: mSystems00954-24 (Different artificial feeding strategies shape the diverse gut microbial communities and functions with the potential risk of pathogen transmission to captive Asian small-clawed otters (*Aonyx cinereus*))

Dear Dr. Long Guo:

Revision Guidelines

Sincerely,
Suzanne Ishaq
Editor
mSystems

Reviewer #1 (Comments for the Author):

In this manuscript, the authors detected the microbiota in feces from otters with metagenomic sequencing and analyzed the microbial functions. This article focuses on the impact of artificial feeding strategies for captive otters on their gut microbiota and also discovers the enrichment of pathogenic microorganisms and resistance genes in otter gut microbiota, thereby pointing out potential problems in the current captive otter feeding process. This work provides new insight and opinion into the gut

microbiota of captive otters under different feeding strategies and is very worthy of study and discussion. The manuscript is well-organized and clearly stated. However, there are still some shortcomings in this article.

1. The section on "Importance" at the beginning of the manuscript does not provide an accurate and sufficient summary of the main points studied and discussed in the article. There are certain linguistic errors and grammatical mistakes in this part, such as the mistakes in tense and sentence subject in the first sentence, which are lines 42 to 44 of the manuscript.
2. The content from lines 52 to 56 of the article appears somewhat redundant and unnecessary for the first paragraph of the introduction and can be directly deleted or streamlined. The other part of the "Introduction" section of the article regarding the research background can be appropriately modified and refined to ensure a more accurate and direct presentation of your research objectives.
3. In the discussion on the enrichment of resistance genes (lines 571-573), it was mentioned that the raw loaches supplemented with commercial cat food (LSCF) diet are high in protein and fat. However, I believe this is only compared to another diet, otherwise, a clear definition of high protein or a high-fat diet should be established. Therefore, the discussion here should be more rigorous in its wording.
4. In the description of microbial composition, functional composition, and differences in the "Results" section, unnecessary detailed descriptions of the results can be appropriately reduced, while emphasizing and presenting the differences between the two groups and the connections between the results in each section during the discussion.
5. In the discussion section of the article, the author believes that the enrichment of various pathogenic bacteria in feces may come from the food of otters. That may not be rigorous, so the authors need to dig deeper.
6. In line 138 of the article, in the Materials and Methods section, there are two font settings for the letter 's' that need to be adjusted.
7. The citation format of some references in the article is not very standardized, such as paying attention to the spelling of journal names.
8. The layout between the sub-images in the combination figures of results in the article is somewhat dense and not very neat. It is possible to consider splitting or reassembling the images appropriately, such as Figure 5.
9. In Figure 7A, it is recommended not to use mmol/L to represent the content of SCFAs, but to use mmol/g instead. This is because feces are solid and using mmol/L is not appropriate, otherwise, the dilution ratio for extracting SCFAs from feces needs to be clarified.

Reviewer #2 (Comments for the Author):

On line 82, the presentation's logic appears inconsistent with the preceding content. It is assumed that the authors aim to use the example of the reversal effect of commercial dry food on the gut microbiota composition in dogs fed a meat-based diet to emphasize the significant impact of commercial dry food on the gut microbiota composition.

In Figure 5, parts of B and E are missing confidence intervals for group B. The confidence intervals for group B are missing. To ensure the consistency of the figure, confidence intervals should be added for group B or deleted for the corresponding part of group A.

I note that the use of 'Group B' and 'LSCF diet group' in the article is confusing, so please standardise the presentation.

Nutritional measurements were performed in the article, and I suggest that the authors provide a systematic and detailed description of the nutritional measurements in the 'Results' section.

**Different artificial feeding strategies shape the diverse gut microbial**
**communities and functions with the potential risk of pathogen**
**transmission to captive Asian small-clawed otters (*Aonyx cinereus*)**

Yuanda Gao ^a, Hangyu Zhang ^a, Dapeng Zhu ^{b*}, Long Guo ^{a*}

7 ^aState Key Laboratory of Herbage Improvement and Grassland Agro-Ecosystems,
College of Pastoral Agriculture Science and Technology, Lanzhou University,
Lanzhou, 730020, China.

10 ^bFoping National Nature Reserve, Hanzhong 723000, China

* Corresponding author.

E-mail address: guolong@lzu.edu.cn (L. Guo)

**Abstract**

The objective was to evaluate how impacts of two different artificial diets on
microbial communities and function capabilities and short-chain fatty acids (SCFAs)
profiles in healthy otters' feces. A total of 16 Asian small-clawed otters in two groups
(n=8) were selected, group A was a raw crucian (RC) diet and group B was raw
loaches supplemented with commercial cat food (LSCF). The living conditions of the
two groups of animals were consistent. The communities and functional capabilities
of microbiota in the feces were assessed with metagenomic sequencing. Intestinal
metabolic activity was determined by analysis of SCFAs in feces. Captive otters fed
two kinds of diets possessed different gut microbial communities and functional
capabilities. Various pathogenic bacteria, like *Escherichia_coli* and *Clostridium*
*perfringens*, were enriched in the samples from two groups, respectively. Most of the
differential pathways of nutrient metabolism were significantly enriched in the LSCF
diet group, and the distributions of carbohydrate enzymes in the two groups
significantly differed from each other. Multiple resistance genes markedly
accumulated in fecal samples of the LSCF diet otters. Higher concentrations of
SCFAs were also observed in otters consuming LSCF diet. Two feeding strategies
were both likely to facilitate the colonization and expansion of various pathogenic
bacteria and the accumulation of resistance genes in the intestines of captive otters,
suggesting that risk of pathogen transmission existed in the current feeding process.
Commercial cat food could supplement various nutrients and provide a substrate for
the production of SCFAs, which might be beneficial for the otter's health.

**Keywords:** Asian small-clawed otter, Diet, Gut microbiota, Metagenomic
sequencing, Short chain fatty acids

**Importance**

Captive otters with different diet possessed distinct gut microbial communities
and functions, while accelerate the colonization and proliferation of various
pathogenic bacteria in the intestines of otters and the spread of resistance genes,
increasing risk of diseases. Diverse food sources had benefits for otters' intestinal
fermentation and metabolism of gut microbiota, and the enrichment of several
pathogens and multiple resistance genes due to the possible shortcomings in current
feeding strategies.

INTRODUCTION

Otter belongs to the Mustelidae family of the Carnivora order, a semi-aquatic
carnivore and a top predator living in freshwater aquatic ecosystems, playing a vital
role in maintaining ecosystem functions [1, 2]. There are 13 species and 7 genera of
otter in the world [3], three species of otter are known from China: the Eurasian otter,
the Asian small-clawed otter (*Aonyx cinereus*), and the smooth-coated otter *Lutrogale*
*perspicillata* [4]. The Asian small-clawed otter is a species of the genus of *Aonyx*, a
second-class protected animal in China, distributed in southeast Asian countries, India
and several southern provinces of China, and it inhabits rivers, streams and shallow
waters along estuaries in the wild [5]. Except for the otters distributed in the wild,
there are also a certain number of otters kept in captivity at present for ex-situ
conservation, offering a chance for humans to understand the living habits and
characteristics of otters.

The sources and types of the diets of wild otters are diverse and variable. Fish is
the main prey category in the diet of wild otters who also prey on clams, crabs,
crayfish, amphibians, insects and various small mammals when fish availability
decreases [6-8]. However, unlike wild otters, the diet of captive otters is relatively
single and affected by feeding management and also might not be suitable and
beneficial for the otter's health. Previous studies on various captive animals have
revealed that diet ratio was one of the main causes of several diseases occurring in
captive animals. The diet of polar bears in zoos had a higher protein content compared
to the diet in the wild, which may be a contributing factor to the end-stage renal

disease and cardiovascular disease occurring in captive polar bears [9, 10]; The
investigation on Vancouver Island marmots (*Marmota vancouverensis*) indicated that
artificial diet induced captive marmots to have higher adipose tissue reserves, leading
in turn to cardiovascular disease, the primary cause of mortality in captive marmots
[11]. At present, otters in various captive conditions or zoos are often given different
diets, among which the nutrition supply may vary greatly. Substantial prior studies
have documented that diet is an important external factor deeply affecting the makeup
and diversity of gut microbiota [12, 13]. As a result, the dietary difference of captive
otter might lead to differential gut microbial flora. Moreover, some captive otters have
been supplementarily fed commercial cat or dog food containing starch, dietary fiber,
vitamins, minerals, and other plentiful nutrient substances with the fish or meat-based
diet. We lack a clear understanding of the impact of feeding dry commercial pet food
on the health and gut microbiota of otters. A previous study on canines indicated
meat-based diet seemed to induce changes in fecal microbiota composition compared
with that in dogs fed commercial dry food [14]. Consequently, a variation of the
microbial community might have occurred in the intestine of these captive otters fed
pet food.

Gut microbiota is the microbial communities colonized in all parts of animal
intestines, composed of a diverse range of symbiotic gut bacteria and other massive
microorganisms [15]. Apart from playing a crucially important role in nutrient
digestion and metabolism [16, 17], it has been proven to be linked to the onset and
development of numerous diseases and involved in the modulation of host health

depending on multiple microbiota-derived metabolites [18, 19], such as short chain
fatty acids (SCFAs) [20]. Additionally, the alteration of gut microbiota can be
associated with some diseases in humans [21, 22], including obesity [23], type 2
diabetes [24], nonalcoholic fatty liver disease (NAFLD) [25], and inflammatory
bowel disease (IBD) [26]. We, therefore, believe that the various artificial diets could
have different potential effects on the health and growth of captive otters by
influencing their gut microbiota.

Currently, several researches have been conducted on wild otters, investigating
and answering a few questions regarding the habitat protection, population status and
diets of wild otters [27]. However, to the best of our knowledge, there remains a
scarcity of research on Asian small-clawed otters under captive conditions, including
their diet and gut microbiota, although there are also a few studies about gut
microbiota have been carried out using other otter species [28, 29]. Accordingly, in
this work, we selected two groups of captive Asian small-clawed otters with distinct
dietary patterns from two representative zoos in China and collected their feces.
Metagenomic sequencing was performed to characterize and compare the structures,
compositions and functions of gut microbial communities in two groups. This trial
was undertaken to identify the impact of different feeding strategies and nutrient
supply on gut microbiota in captive otters, to better understand the variability of gut
microbial ecosystem under different feeding conditions and to explore the potential
effect of artificial diet on otters' health. This could be essential and directive for
improving their feeding management and ex-situ conservation strategies in the future.

MATERIALS AND METHODS

**Animals, diets and experimental design**

Sixteen healthy Asian small-clawed otters (*Aonyx cinereus*) with average age of
6-year-old were selected in this study. Otters from two groups were given two
completely different kinds of foods respectively: group A otters were fed raw loaches
(*Misgurnus anguillicaudatus*) mainly, supplemented with some commercial cat food
(LSCF diet; loaches: 84%, commercial cat food: 16%), 600 g per day; group B otters
were fed raw crucians (*Carassius auratus*) (RC diet), 600 g per day. Otters in two
group were raised in the same environment and were given the same management.

**Feed sampling and analysis**

The feed samples of two groups were randomly collected from the normal diet of
otters and then stored at -20°C until nutrient contents analysis. Crucian and loach
samples were freeze-dried at -60°C for 76 h, and cat food samples were first dried in
a forced-air oven at 65°C for 48 h. Dry matter (DM), crude protein (CP), ether extract
(EE) also called crude fat (CF) and ash content of feed samples were measured as
described previously [63]. The gross energy (GE) content of the dried samples was
analyzed by combustion in an adiabatic bomb calorimeter (IKA C3000, Germany).
The nutrient compositions and contents of diets were detected and presented in Table
1, and the otters' intakes of diet and several nutrients in two groups were shown in
Table 2.

**Fecal Sample collection**

Fresh fecal samples of otters in the two places were collected and stored with

sterilized fecal collection tubes without contamination and then transported to the
laboratory immediately with dry ice. These samples were stored at -80°C until DNA
extraction. All samples were collected from April to June in 2022.

**DNA extraction, library construction, and metagenomic sequencing**

All fecal samples were thawed at 4°C before DNA extraction, 200 mg per fecal
sample was used for microbial DNA extraction using the E.Z.N.A.® Stool DNA Kit
(Omega Bio-tek, Norcross, GA, U.S.) according to manufacturer's instructions
according to the manufacturer's protocol. The concentration, integrity and purity of
DNA samples were determined using a Nanodrop 2000 spectrophotometer and 1%
agarose gel electrophoresis. DNA extract was fragmented to an average size of about
400 bp using Covaris M220 (Gene Company Limited, China) for paired-end library
construction. Paired-end library was constructed using NEXTflex™ Rapid DNA-Seq
(Bioo Scientific, Austin, TX, USA). Adapters containing the full complement of
sequencing primer hybridization sites were ligated to the blunt-end of fragments.
Paired-end reads of metagenomic libraries were sequenced on an Illumina Novaseq
6000 (Illumina Inc., San Diego, CA, USA) at Wefindbio Technology Co., Ltd.
(Wuhan, China) using NovaSeq Reagent Kits according to the manufacturer's
instructions.

**Sequence quality control, gene prediction, and genome assembly**

The quality control of raw data was performed to trim sequencing adapters,
filtering out low-quality reads (reads with N bases, quality scores < 20) and short
reads (<50 bp) by using fastp (<https://github.com/OpenGene/fastp>, v.0.20.0) [64], and

BWA (<http://bio-bwa.sourceforge.net>, v.0.7.9a) was used to filter out the reads
mapped to the host originated genes to obtain high-quality reads [65]. A total of 164
163 Gb high quality and clean reads of each sample obtained after quality control were
164 assembled into contigs processed by Megahit (parameters: kmer_min=47,
kmer_max=97, step=10) (<https://github.com/voutcn/megahit>, v.1.1.2) [66], which
makes use of succinct de Bruijn graphs.

Contigs with a length exceeding 800 bp were obtained finally and used for
subsequent analysis. Afterward, MetaGene (<http://metagene.cb.k.u-tokyo.ac.jp/>) [67]
was applied to predict open reading frames (ORFs) of these contigs. The predicted
ORFs with lengths being or over 100 bp were retrieved and translated into amino acid
sequences using the NCBI translation table. All the predicted ORFs were combined to
generate the nonredundant microbial gene catalog using CD-HIT
(<http://www.bioinformatics.org/cd-hit/>, v.4.6.1) [68], with clustering criteria of $\geq 95\%$
identity and $\geq 90\%$ overlap. Then, all clean reads of each sample were aligned to the
nonredundant gene catalog using SOAPaligner (<http://soap.genomics.org.cn/>, v.2.21)
[69] with the criteria of $\geq 95\%$ identity to get the specific gene abundance information
in samples. We finally obtained 2940768 nonredundant genes with the 489.4 bp
average length and the 579 bp N50 length.

Genome reconstruction or genome binning of gut microbes with metagenomic
sequences was carried out by Vamb (<https://github.com/RasmussenLab/vamb/>,
v.3.0.5) [70] for assembling contigs from each single sample into metagenomic bins.
dRep (<https://drep.readthedocs.io/>, v.3.0.0) [71] was used to filter the replications of

all bins. The completeness and contamination of all bins were estimated by CheckM
(<https://github.com/Ecogenomics/CheckM/>, v.1.1.3) [72] according to the quality
evaluation criteria (more than 50% completeness and 10% contamination), and a total
of 160 nonredundant bins were identified as metagenome-assembled genomes
(MAGs) for downstream analysis. All MAGs were annotated with a taxonomy using
GTDB-Tk (v.1.7.0) [73] based on the Genome Taxonomy Database
(<https://gtdb.ecogenomic.org/>).

**Taxonomy and Function annotation of genes**

To gain insight into the taxonomy profiles of the metagenome of gut microbiota,
all the representative sequences of nonredundant genes were aligned to the sequences
belonging to bacteria, fungi, archaea and viruses in the NR database (v.2021.11) using
blastp with e-value cutoff of $1e-5$ using DIAMOND (<http://www.diamondsearch.org/>,
v.0.8.35) [74]. The genes that could not be classified into any taxa were defined as
unknown taxa. The KEGG and cluster of orthologous groups of proteins (COG)
annotation for the representative sequences were performed respectively by aligning
genes to the Kyoto Encyclopedia of Genes and Genomes (KEGG) database
(<http://www.genome.jp/kegg/>, v.94.2) [75] with an e-value cutoff of $1e-5$ and eggNOG
4.5.1 database [76] using DIAMOND (v.0.8.35) [74]. Carbohydrate-active enzymes
(CAZymes) were annotated by aligning genes to the CAZyme database
(<http://www.cazy.org/>) [77] using hmmscan (<http://hmmer.org/>) with an e-value cutoff
of $1e-5$. Antibiotic resistance genes (ARGs) were annotated against Antibiotic
Resistance Genes Database (ARDB) (<http://ardb.cbcb.umd.edu/>, v.1.1) and the

Comprehensive Antibiotic Research Database (CARD)
(<https://card.mcmaster.ca/home>, v.3.0.9) [78] also with DIAMOND (v.0.8.35) [74]
through aligning unigenes as above mentioned with $evalue \leq 1e-30$. Differential
functional categories and resistance genes were identified by STAMP (v 2.1.3) [79].

**SCFAs analysis of feces**

All samples were diluted with distilled water and then centrifuged at 2,000 g for
10 min to separate the solid residues and liquid. The supernatant fluid was mixed with
25% (wt/vol) metaphosphoric acid (5 ml supernatant fluid and 1 ml metaphosphoric
acid) and stored at -20°C .) Concentrations of SCFAs (Acetate, Propionate,
Isobutyrate, Butyrate, Isovalerate) were qualified by gas chromatography (Thermo
Fisher Scientific, USA) at the College of Pastoral Agriculture Science and
Technology, Lanzhou University.

**Cooccurrence network analysis**

The correlation network within group A and B otters was calculated separately
by Spearman's correlation coefficient with the R package Spaa (v.0.2.2), to reduce
network complexity, those species total abundance > 1 in all samples and presented in
at least all 8 samples were used for Cooccurrence network analysis. For the analysis
of MAGs, all MAGs obtained were used for co-occurrence network construction.
Spearman's correlation coefficient between two MAGs was considered robust if the
absolute r-value > 0.6 with a corresponding 'fdr' adjusted p-value < 0.05 and the
significant and robust correlations between species were defined as absolute r-value $>$
0.8 with a corresponding 'fdr' adjusted p-value < 0.05 . Co-occurrence modules were

analyzed by R package igraph (v.1.4.1) with 9 modules presented in each group.
These correlations obtained above were graphed using Gephi (v.0.10.1) [80] with the
layout algorithm of Fruchterman Reingold.

**Statistical analysis**

SPSS 27 was used for the statistical analysis and R (v.4.2.2) was used for the
visualization of all processed data unless otherwise stated. For clustering heatmaps,
the data were normalized using z-scores of the abundance of the top 20 bacteria taxa
and were visualized by the Pheatmap package in R language. LEfSe analysis of
differential bacteria taxa was performed using the microeco R package (1.3.0) for data
processing, analysis, and plotting.

**RESULTS**

**Overview of samples collected, metagenomic sequencing data, and gene catalogs**

The metagenomic sequencing data from 16 Asian small-clawed otters fecal
samples were sequenced in this study, including 8 otters fed raw loaches
supplemented with commercial cat food (LSCF) diet in group A and the other 8 otters
fed raw crucian (RC) diet in group B. Metagenomic sequencing of DNA samples
generated high-quality clean data after removing low-quality reads and host reads.
After de novo assembly, gene prediction and filtering of incomplete genes, 2,940,768
complete unique genes were identified. A total of 13354 core genes coexisted in all
samples and sample B_2 had the highest number of genes with 1,494,638 unique
genes (Figure 1A). Furthermore, Spearman correlation analysis of gene abundance
patterns between groups A and B was carried out, indicating that higher similarity of

gene abundance patterns was among the same group samples, and the correlations of
samples from different groups were lower (Figure 1B).

**Taxonomic characteristics of gut microbial communities**

To investigate the effects of two diets on the gut microbial taxonomic
characteristics of otters, the distribution, composition and fluctuations of microbiota
in feces samples obtained from 16 otters were identified by metagenomic sequencing.
Matching the metagenomic genes to the NR (Non-Redundant Protein Sequence
Database) database of NCBI (National Center of Biotechnology Information) for
taxonomic annotation, we found that more than 99.05% and 73.22% of the classified
genes were assigned to Bacteria in group A and B respectively, only 1.1% and 26.78%
of the remaining genes belonged to Eukaryota, Archaea and Viruses in two groups
(Figure 1C).

The relative abundance of the different bacteria at all phylogenetic levels was
identified among the two groups of otters fed with different diets, and the top 20
bacterial taxa of genus and species in abundance were depicted in Figure 2A-2B and
Figure 3A-3B. 438 core genera were found distributed in all 16 samples (Figure 2F).
The most major taxa identified at the genus level in the feces samples from group A
was *Clostridium* (32.52% ± 10.87%), followed by *Aeromonas* (22.99% ± 10.69%),
*Escherichia* (8.69% ± 8.82%), *Cetobacterium* (7.27% ± 7.99%), *Plesiomonas* (4.63%
± 1.77%), while *Clostridium* (61.56% ± 24.66%), *Romboutsia* (8.64% ± 4.29%),
*Aeromonas* (2.41% ± 4.27%), *Escherichia* (0.14% ± 0.19%) were the 5 most
prevalent genera in group B (Figure 2A and 2B). At the species level, we identified

different numbers of bacteria species in the fecal samples from two groups and also
found 472 core species shared in all samples with Venn diagram analysis (Figure 3F).
Furthermore, the bacterial communities in the gut of otters from two groups were
dominated by several main species such as *Clostridium_perfringens* (47.69% ±
16.74% for A; 69.12% ± 25.12% for B), *Escherichia_coli* (13.38% ±13.79% for A;
0.18% ± 0.24% for B) and *Cetobacterium_somerae* (3.56% ± 3.82% for A; 0.22% ±
0.39% for B) (Figure 3A and 3B). Although these species were all believed to be
pathogenic bacteria associated with a variety of diseases, the above results in this part
indicated that the two group's otters possessed distinct relative abundances and
compositions of main gut bacteria.

**Discrepancies in gut microbial communities between two groups**

Significantly higher values of alpha index were observed in group A compared to
group B in the estimations of alpha diversity at genus and species level (Figure 2D
and Figure 3D). A Higher Simpson index was observed in group A at the two
taxonomy levels, and a higher Richness index was observed in group B at the genus
level. Principal coordinates analysis (PCoA) based on Bray-Curtis with an analysis of
similarities (Anosim) was performed to characterize the divergence of the gut
microbial community of two groups at genus and species level as well (Figure 2E and
Figure 3E). PCoA score plots for two groups showed a clear separation that bacterial
communities from one group clustered together and separated from those in the other
group, suggesting that the inter-group differences between two groups were
significantly greater than the intra-group differences ($r = 0.352$, $P = 0.005$, at species

level) and that two groups of otters harbored distinct bacterial floras. Clustering
analysis was used to further determine the similarities and differences in the top 20
microbial species abundances between samples (Figure 3H).

To further explore the differential gut bacteria between the two groups, we
performed linear discriminant analysis effect size (LEfSe) to screen significantly
different biomarkers discriminating the two groups. Using more stringent criteria
(LDA score > 3.5 and KW test P -value < 0.001), 18 bacterial genera and species were
identified as significantly differentially represented between the two groups
respectively (Figure 2C and Figure 3C). At the species level, 11 bacterial species were
significantly enriched in group A, and group B was distinguished from the other 7
bacteria (Figure 3C). For instance, *Escherichia_coli* was notably enriched in group A
but depleted in group B, whereas *Clostridium_perfringens* was more prevalent in
group B but declined in group A.

We constructed a co-occurrence network analysis at the species level to
characterize the interactions of gut microbiota in each group. The correlation
networks were analyzed by Spearman's correlation coefficient and the significant
correlations were visualized as Figure 3G. These species of gut microbiota in two
groups were clustered into 9 modules respectively. The correlation network of group
B (nodes:260, edges:4831) possessed more nodes and edges than group A (nodes:229,
edges:2302) at species level, filtered in stricter threshold ($P < 0.05$, $|\rho| > 0.80$). The
correlations between nodes and modules in the group A species co-occurrence
network are less tight than that in group B, especially module 2 in group A nearly

completely separated from other modules without any strong correlation. Overall, the
result of the co-occurrence network analysis demonstrated that the gut bacteria of
group B connected more closely, suggesting the gut microbial community of group B
was more stable than that of group A.

**Reconstruction of microbial metagenomic genomes**

Metagenomic contigs from 16 fecal samples were reassembled and clustered into
160 microbial genomes belonging to strains with a threshold of >50% completeness
and contamination of $\leq 10\%$ (Figure 4I). Then we used the NT (Nucleotide Sequence
Database) database of NCBI for taxa annotation of these reconstructed genomes, 160
MAGs were classified into Bacteria and Archaea kingdom. Additionally, 147 of these
MAGs were annotated to 8 different phyla, 113 MAGs were aligned to 34 genera, but
only 60 MAGs were annotated to species level (Figure 4A-4D). We presented the
compositions and relative abundances of the top 20 MAGs belonging to 4 taxa within
each sample as shown in Figure 4E, meantime the analysis of the top 20 MAGs'
average relative abundances in two groups was shown in Figure 4F, indicating the
distribution patterns of these MAGs in two groups were significantly different (Figure
4G). A co-occurrence network analysis of MAGs in two groups was also performed,
showing a result that the MAGs in group B connected more tightly and more stable,
similar to the result of previous analysis at the species taxonomic level (Figure 4H).
STAMP analysis revealed several MAGs possessed different abundances in the two
groups. 5 MAGs like "S15C2549" and "S14C2549" were differentially enriched in
group A. All of the 5 MAGs belonged to *c_Gammaproteobacteria* which holds more

than 20 genera containing some taxa that infect humans and animals [30], such as the
bacterium *Escherichia coli*, well-known pathogens *Salmonella*, *Yersinia*, *Vibrio*, and
*Pseudomonas* [31]. Meanwhile, there were also other 5 MAGs markedly existing in
group B. Four of them were annotated to *f-Clostridiaceae* and the other MAG,
“S4C5569”, was aligned to *s_Clostridium perfringens* (Figure 4J).

**Functional characteristics and differences of gut microbial communities in otters** 343 **from two groups**

To characterize the functions of the gut microbiota of these captive otters,
metagenomic genes were aligned to the Kyoto Encyclopedia of Genes and Genomes
(KEGG), evolutionary genealogy of genes: non-supervised orthologous groups
(eggNOG) and Carbohydrate-active Enzymes (CAZymes) databases. The KEGG
enrichment analysis of metagenomic genes from all samples confirmed 438 third-
level pathways, in which 388 pathways were shared in two groups. All of these
pathways obtained belonged to 6 first-level categories and 45 second-level categories
(Figure 5A). “Metabolism” was found to be the most predominant first-level pathway
in two groups (Figure 5A), and “Metabolism of cofactors and vitamins”, “Membrane
transport”, and “Carbohydrate metabolism” were the most dominant 3 second-level
categories, followed by “Replication and repair”, and “Energy metabolism” (Figure
5A), highlighting the importance of these metabolic pathways in gut microbiota. At
the third level, the relative abundances of 20 top pathways were visualized (Figure
5C), meanwhile, PCoA analysis of KEGG level 3 presented a noted separation (Figure
5B). In addition, eggNOG functional annotation identified 24 categories in level 1

with “Replication, recombination, and repair” being the most abundant in all
categories except “Function unknown” (Figure 5D). Functions connected with the
transport and metabolism of nutrients were also highly abundant, such as “Energy
production and conversion”, “Amino acid transport and metabolism”, “Lipid transport
and metabolism” and “Carbohydrate transport and metabolism”. Similarly, the
separation of distribution patterns and the differences in relative abundances of
eggNOG categories of the two groups were verified as well (Figure 5E and 5F). These
results indicated that the microbial functions of the two groups’ samples differed from
each other in annotations of the KEGG and eggNOG databases.

Subsequently, STAMP (Statistical analysis of metagenomic profiles) was used
for the identification of functions with significant differences between groups. The
significantly differential KEGG pathways are shown in Figure 5G. There were 31
KEGG functional pathways with marked enrichment in group A and 9 pathways
enriched in group B. For instance, “Lipopolysaccharide biosynthesis” correlated with
the production of gram-negative bacteria, and some pathways related to the
production of opportunistic pathogens and diseases including “Biofilm formation -
*Escherichia coli*” and “Biofilm formation-*Vibrio cholerae*” were all enriched in group
376 A. Whereas, pathways like “Thyroid hormone signaling pathway” and “Nucleotide
excision repair” were enriched in group B (Figure 5G). Additionally, we also
discovered that functions involving “Inorganic ion transport and metabolism” and
“Amino acid transport and metabolism” were remarkably abundant in group A, while
“Nuclear structure” and “RNA processing and modification” were the most active

eggNOG functional categories in group B (Figure 5H).

For CAZyme profiles, 79446 CAZyme families were identified as belonging to 6
functions categories with the annotation of the CAZy database, in which the GH
(Glycoside Hydrolase) and GT (GlycosylTransferase) were the most annotated 2
categories (Figure 6A). The relative abundances of the top 20 CAZyme families are
presented in Figure 6B. The PCoA analysis exhibited a considerable difference
between the two groups (Figure 6C). Additionally, STAMP analysis showed that
GH23 and CE1 were the most significantly enriched CAZymes in group A (Figure
6G). Furthermore, the total relative abundance of GH13 CAZyme in two groups was
focused on its specific function in starch degradation, showing that the relative
abundance of GH13 was significantly higher in group A (Figure 6D).

Furthermore, Metagenomic genes were also classified to the Comprehensive
Antibiotic Research Database (CARD) to annotate the antibiotic resistance genes
(ARGs) that existed in otters' gut microbiota. We identified 232 different Antibiotic
Resistance Ontologies (AROs) in all samples, with significantly different abundances
in all samples and most AROs significantly enriched in group A (Figure 6E and 6F).
The most abundant 10 AROs conferring resistance to 8 groups of antibiotics
comprising macrolides, elfamycins, aminoglycosides, β -lactams, tetracyclines,
quinolones, sulfonamides and polypeptides.

**The associations between SCFAs profiles and differential gut bacteria taxa**

[revised manuscript text omitted]

Feeding raw fish or other raw aquatic products would induce gut microbiota
dysbiosis [40] and might increase the risk of gastrointestinal diseases [41], system and
chronic diseases [42] by the transmission of numerous pathogenic bacteria from feed
to otters. As claimed by LEfSe, several opportunistic pathogenic bacteria species, like
*Escherichia_coli*, *Aeromonas veronii*, and *Aeromonas hydrophila*, were significantly
enriched in the LSCF diet group, in accordance with the result of MAGs analysis. As
we all know, *Escherichia_coli* is an opportunistic pathogen and an important member
of the normal intestinal bacteria of humans or other animals, which can cause
intestinal diseases and extra-intestinal infections in various parts of the body when the
host is immunocompromised [43]. *Aeromonas veronii* is a highly pathogenic bacteria
with a wide range of hosts, widely exists in the environment of humans, animals, and
aquatic animals, and can cause a variety of diseases [44]. The bacteria *Aeromonas*

*hydrophila* is a zoonotic bacterial pathogen, that frequently causes disease and mass
mortalities among cultured and feral fishes [45]. Additionally, two other pathogenic
bacteria, *Yersinia ruckeri* and *Plesiomonas shigelloides*, were enriched in the LSCF
diet group as well. *Yersinia ruckeri* is a Gram-negative rod-shaped enterobacterium
and the causative agent of enteric redmouth disease (ERM) affecting salmonids and
other commercial fish species in the world [46]. *Plesiomonas shigelloides* is a Gram-
negative bacillus belonging to the Enterobacteriaceae family and will cause

[revised manuscript text omitted]

Archive (SRA) database under the accession number PRJNA1105391.

**References**

- 1. Shekhar Palei H, Mohapatra PP, Hussain SA. Habitat selection and diet of the
Asian small-clawed otter in Karlapat Wildlife Sanctuary, Odisha, India.
*Écoscience*. 2023;30(1):17-26; doi: 10.1080/11956860.2023.2165020.
- 2. Zhang R, Yang L, Laguardia A, Jiang Z, Huang M, Lv J, et al. Historical
distribution of the otter (*Lutra lutra*) in north-east China according to historical
records (1950-2014). *Aquatic Conservation: Marine and Freshwater
Ecosystems*. 2016;26(3):602-6; doi: 10.1002/aqc.2624.
- 3. Langer P. *Mammal Species of the World: A Taxonomic and Geographic
Reference*. *Mammalian Biology*. 2007;72(3):191-; doi:
10.1016/j.mambio.2006.02.003.
- 4. Li F, Chan BPL. Past and present: the status and distribution of otters
(*Carnivora: Lutrinae*) in China. *Oryx*. 2017;52(4):619-26; doi:
10.1017/s0030605317000400.
- 5. Jiang Z, Ma Y, Wu Y, Wang Y, Zhou K, Liu S, et al. China's mammal diversity
and geographic distribution. Beijing: Science Press; 2015.

- 6. Remonti L, Prigioni C, Balestrieri A, Sgrosso S, Priore G. Trophic flexibility
of the otter (*Lutra lutra*) in southern Italy. *Mammalian Biology*.
2008;73(4):293-302; doi: 10.1016/j.mambio.2007.04.004.
- 7. Krawczyk AJ, Bogdziewicz M, Majkowska K, Glazaczow A. Diet
composition of the Eurasian otter *Lutra lutra* in different freshwater habitats of
temperate Europe: a review and meta-analysis. *Mammal Review*.
2016;46(2):106-13; doi: <https://doi.org/10.1111/mam.12054>.
- 8. Beja PR. An Analysis of Otter *Lutra lutra* Predation on Introduced American
Crayfish *Procambarus clarkii* in Iberian Streams. *Journal of Applied Ecology*.
1996;33(5):1156-70; doi: 10.2307/2404695.
- 9. Robbins CT, Tollefson TN, Rode KD, Erlenbach JA, Arden AJ. New insights
into dietary management of polar bears (*Ursus maritimus*) and brown bears
(*U. arctos*). *Zoo Biol*. 2022;41(2):166-75; doi: 10.1002/zoo.21658.
- 10. LaDouceur EEB, Garner MM, Davis B, Tseng F. A RETROSPECTIVE
STUDY OF END-STAGE RENAL DISEASE IN CAPTIVE POLAR BEARS
(*URSUS MARITIMUS*). *Journal of Zoo and Wildlife Medicine*.
2014;45(1):69-77; doi: 10.1638/2013-0071r.1.
- 11. Aymen J, Delnatte P, Beaufriere H, Chalil D, Steckel KE, Gourlie S, et al.
Comparison of blood leptin and vitamin E and blood and adipose fatty acid
compositions in wild and captive populations of critically endangered
Vancouver Island marmots (*Marmota vancouverensis*). *Zoo Biol*.
2023;42(2):308-21; doi: 10.1002/zoo.21739.

- 12. David LA, Maurice CF, Carmody RN, Gootenberg DB, Button JE, Wolfe BE,
et al. Diet rapidly and reproducibly alters the human gut microbiome. *Nature*.
2014;505(7484):559-63; doi: 10.1038/nature12820.
- 13. Gentile CL, Weir TL. The gut microbiota at the intersection of diet and human
health. *Science*. 2018;362(6416):776-80; doi: doi:10.1126/science.aau5812.
- 14. Herstad KMV, Gajardo K, Bakke AM, Moe L, Ludvigsen J, Rudi K, et al. A
diet change from dry food to beef induces reversible changes on the faecal
microbiota in healthy, adult client-owned dogs. *BMC Veterinary Research*.
2017;13(1); doi: 10.1186/s12917-017-1073-9.
- 15. Nicholson JK, Holmes E, Kinross J, Burcelin R, Gibson G, Jia W, et al. Host-
Gut Microbiota Metabolic Interactions. *Science*. 2012;336(6086):1262-7; doi:
doi:10.1126/science.1223813.
- 16. Rowland I, Gibson G, Heinken A, Scott K, Swann J, Thiele I, et al. Gut
microbiota functions: metabolism of nutrients and other food components. *Eur*
*J Nutr*. 2018;57(1):1-24; doi: 10.1007/s00394-017-1445-8.
- 17. Oliphant K, Allen-Vercoe E. Macronutrient metabolism by the human gut
microbiome: major fermentation by-products and their impact on host health.
*Microbiome*. 2019;7(1):91; doi: 10.1186/s40168-019-0704-8.
- 18. de Vos WM, Tilg H, Van Hul M, Cani PD. Gut microbiome and health:
mechanistic insights. *Gut*. 2022;71(5):1020-32; doi: 10.1136/gutjnl-2021-
326789.
- 19. Wu J, Wang K, Wang X, Pang Y, Jiang C. The role of the gut microbiome and

- its metabolites in metabolic diseases. *Protein Cell*. 2021;12(5):360-73; doi:
10.1007/s13238-020-00814-7.
- 20. Martin-Gallausiaux C, Marinelli L, Blottiere HM, Larraufie P, Lapaque N.
SCFA: mechanisms and functional importance in the gut. *Proc Nutr Soc*.
2021;80(1):37-49; doi: 10.1017/S0029665120006916.
- 21. Beam A, Clinger E, Hao L. Effect of Diet and Dietary Components on the
Composition of the Gut Microbiota. *Nutrients*. 2021;13(8); doi:
10.3390/nu13082795.
- 22. Fan Y, Pedersen O. Gut microbiota in human metabolic health and disease.
*Nat Rev Microbiol*. 2021;19(1):55-71; doi: 10.1038/s41579-020-0433-9.
- 23. Turnbaugh PJ, Ley RE, Mahowald MA, Magrini V, Mardis ER, Gordon JI. An
obesity-associated gut microbiome with increased capacity for energy harvest.
*Nature*. 2006;444(7122):1027-31; doi: 10.1038/nature05414.
- 24. Qin J, Li Y, Cai Z, Li S, Zhu J, Zhang F, et al. A metagenome-wide association
study of gut microbiota in type 2 diabetes. *Nature*. 2012;490(7418):55-60; doi:
10.1038/nature11450.
- 25. Wang B, Jiang X, Cao M, Ge J, Bao Q, Tang L, et al. Altered Fecal Microbiota
Correlates with Liver Biochemistry in Nonobese Patients with Non-alcoholic
Fatty Liver Disease. *Scientific Reports*. 2016;6(1):32002; doi:
10.1038/srep32002.
- 26. Lavelle A, Sokol H. Gut microbiota-derived metabolites as key actors in
inflammatory bowel disease. *Nat Rev Gastroenterol Hepatol*. 2020;17(4):223-

- 37; doi: 10.1038/s41575-019-0258-z.
- 27. Juarez-Sanchez D, Blake JG, Hellgren EC. Variation in Neotropical river otter
(*Lontra longicaudis*) diet: Effects of an invasive prey species. PLOS ONE.
2019;14(10):e0217727; doi: 10.1371/journal.pone.0217727.
- 28. Guo G, Eccles KM, McMillan M, Thomas PJ, Chan HM, Poulain AJ. The Gut
Microbial Community Structure of the North American River Otter (*Lontra*
*canadensis*) in the Alberta Oil Sands Region in Canada: Relationship with
Local Environmental Variables and Metal Body Burden. Environ Toxicol
Chem. 2020;39(12):2516-26; doi: 10.1002/etc.4876.
- 29. Okamoto Y, Ichinohe N, Woo C, Han S-Y, Kim H-H, Ito S, et al. Contrasting
gut microbiota in captive Eurasian otters (*Lutra lutra*) by age. Archives of
Microbiology. 2021;203(9):5405-16; doi: 10.1007/s00203-021-02526-w.
- 30. Berman JJ. Chapter 7 - Gamma Proteobacteria. In: Berman JJ, editor.
Taxonomic Guide to Infectious Diseases. Boston: Academic Press; 2012. p.
37-47.
- 31. Williams KP, Gillespie JJ, Sobral BW, Nordberg EK, Snyder EE, Shallom JM,
et al. Phylogeny of gammaproteobacteria. J Bacteriol. 2010;192(9):2305-14;
doi: 10.1128/jb.01480-09.
- 32. den Besten G, van Eunen K, Groen AK, Venema K, Reijngoud D-J, Bakker
BM. The role of short-chain fatty acids in the interplay between diet, gut
microbiota, and host energy metabolism. Journal of Lipid Research.
2013;54(9):2325-40; doi: <https://doi.org/10.1194/jlr.R036012>.

- 33. Tan R, Jin M, Shao Y, Yin J, Li H, Chen T, et al. High-sugar, high-fat, and
high-protein diets promote antibiotic resistance gene spreading in the mouse
intestinal microbiota. *Gut Microbes*. 2022;14(1); doi:
10.1080/19490976.2021.2022442.
- 34. Spinler JK, Oezguen N, Runge JK, Luna RA, Karri V, Yang J, et al. Dietary
impact of a plant-derived microRNA on the gut microbiome. *ExRNA*. 2020;2;
doi: 10.1186/s41544-020-00053-2.
- 35. Castañeda S, Ariza G, Rincón-Riveros A, Muñoz M, Ramírez JD. Diet-
induced changes in fecal microbiota composition and diversity in dogs (*Canis*
*lupus familiaris*): A comparative study of BARF-type and commercial diets.
*Comparative Immunology, Microbiology and Infectious Diseases*.
2023;98:102007; doi: <https://doi.org/10.1016/j.cimid.2023.102007>.
- 36. Kim J, An J-U, Kim W, Lee S, Cho S. Differences in the gut microbiota of
dogs (*Canis lupus familiaris*) fed a natural diet or a commercial feed revealed
by the Illumina MiSeq platform. *Gut Pathogens*. 2017;9(1):68; doi:
10.1186/s13099-017-0218-5.
- 37. Butowski CF, Thomas DG, Young W, Cave NJ, McKenzie CM, Rosendale DI,
et al. Addition of plant dietary fibre to a raw red meat high protein, high fat
diet, alters the faecal bacteriome and organic acid profiles of the domestic cat
(*Felis catus*). *PLOS ONE*. 2019;14(5):e0216072; doi:
10.1371/journal.pone.0216072.
- 38. Wardman JF, Bains RK, Rahfeld P, Withers SG. Carbohydrate-active enzymes

- (CAZymes) in the gut microbiome. *Nature Reviews Microbiology*.
2022;20(9):542-56; doi: 10.1038/s41579-022-00712-1.
- 39. Stam MR, Danchin EG, Rancurel C, Coutinho PM, Henrissat B. Dividing the
large glycoside hydrolase family 13 into subfamilies: towards improved
functional annotations of alpha-amylase-related proteins. *Protein Eng Des Sel*.
2006;19(12):555-62; doi: 10.1093/protein/gzl044.
- 40. Shi J, Zhao D, Song S, Zhang M, Zamaratskaia G, Xu X, et al. High-Meat-
Protein High-Fat Diet Induced Dysbiosis of Gut Microbiota and Tryptophan
Metabolism in Wistar Rats. *Journal of Agricultural and Food Chemistry*.
2020;68(23):6333-46; doi: 10.1021/acs.jafc.0c00245.
- 41. Zhang M, Yang X-J. Effects of a high fat diet on intestinal microbiota and
gastrointestinal diseases. *World Journal of Gastroenterology*.
2016;22(40):8905-9; doi: 10.3748/wjg.v22.i40.8905.
- 42. Murphy EA, Velazquez KT, Herbert KM. Influence of high-fat diet on gut
microbiota: a driving force for chronic disease risk. *Current Opinion in*
*Clinical Nutrition & Metabolic Care*. 2015;18(5):515-20; doi:
10.1097/mco.0000000000000209.
- 43. Kaper JB, Nataro JP, Mobley HLT. Pathogenic *Escherichia coli*. *Nature*
*Reviews Microbiology*. 2004;2(2):123-40; doi: 10.1038/nrmicro818.
- 44. Wang Y-d, Gong J-s, Guan Y-c, Zhao Z-l, Cai Y-n, Shan X-f. OmpR (TCS
response regulator) of *Aeromonas veronii* plays a major role in drug
resistance, stress resistance and virulence by regulating biofilm formation.

- Microbial Pathogenesis. 2023;181:106176; doi:
<https://doi.org/10.1016/j.micpath.2023.106176>.
- 45. Kerigano NK, Chibsa TR, Molla YG, Mohammed AA, Tamiru M, Bulto AO,
et al. Phenotypic, molecular detection and antibiogram analysis of *Aeromonas*
*Hydrophila* from *Oreochromis Niloticus* (Nile Tilapia) and Ready-To- eat fish
products in selected Rift Valley lakes of Ethiopia. BMC Veterinary Research.
2023;19(1):120; doi: 10.1186/s12917-023-03684-3.
- 46. Kumar G, Menanteau-Ledouble S, Saleh M, El-Matbouli M. *Yersinia ruckeri*,
the causative agent of enteric redmouth disease in fish. Veterinary Research.
2015;46(1):103; doi: 10.1186/s13567-015-0238-4.
- 47. Cortés-Sánchez ADJ, Espinosa-Chaurand LD, Díaz-Ramirez M, Torres-Ochoa
E. *Plesiomonas*: A Review on Food Safety, Fish-Borne Diseases, and
Tilapia. The Scientific World Journal. 2021;2021:3119958; doi:
10.1155/2021/3119958.
- 48. Ruenkoed S, Wang W. Cloning, characterization, antibacterial activity and
expression of hamp in pond loach (*Misgurnus anguillicaudatus*) after bacterial
challenge with *Aeromonas hydrophila*. Aquaculture. 2019;499:61-71; doi:
<https://doi.org/10.1016/j.aquaculture.2018.09.005>.
- 49. Sandri M, Dal Monego S, Conte G, Sgorlon S, Stefanon B. Raw meat based
diet influences faecal microbiome and end products of fermentation in healthy
dogs. BMC Veterinary Research. 2017;13(1):65; doi: 10.1186/s12917-017-
0981-z.

- 50. Mehdizadeh Gohari I, A. Navarro M, Li J, Shrestha A, Uzal F, A. McClane B.
Pathogenicity and virulence of *Clostridium perfringens*. *Virulence*.
2021;12(1):723-53; doi: 10.1080/21505594.2021.1886777.
- 51. Li X, He L, Luo J, Zheng Y, Zhou Y, Li D, et al. Paeniclostridium sordellii
hemorrhagic toxin targets TMPRSS2 to induce colonic epithelial lesions.
*Nature Communications*. 2022;13(1):4331; doi: 10.1038/s41467-022-31994-x.
- 52. Wolin MJ. Fermentation in the Rumen and Human Large Intestine. *Science*.
1981;213(4515):1463-8; doi: doi:10.1126/science.7280665.
- 53. Verbeke KA, Boobis AR, Chiodini A, Edwards CA, Franck A, Kleerebezem
808 M, et al. Towards microbial fermentation metabolites as markers for health
benefits of prebiotics. *Nutrition Research Reviews*. 2015;28(1):42-66; doi:
10.1017/S0954422415000037.
- 54. Council NR. *Nutrient Requirements of Dogs and Cats*. Washington, DC: The
National Academies Press; 2006.
- 55. Wang A, Zhang Z, Ding Q, Yang Y, Bindelle J, Ran C, et al. Intestinal
*Cetobacterium* and acetate modify glucose homeostasis via parasympathetic
activation in zebrafish. *Gut Microbes*. 2021;13(1):1-15; doi:
10.1080/19490976.2021.1900996.
- 56. Van Boeckel TP, Pires J, Silvester R, Zhao C, Song J, Criscuolo NG, et al.
Global trends in antimicrobial resistance in animals in low- and middle-
income countries. *Science*. 2019;365(6459):eaaw1944; doi:
doi:10.1126/science.aaw1944.

- 57. Liu Y, Liu B, Liu C, Hu Y, Liu C, Li X, et al. Differences in the gut
microbiomes of dogs and wolves: roles of antibiotics and starch. *BMC*
*Veterinary Research*. 2021;17(1); doi: 10.1186/s12917-021-02815-y.
- 58. Tang R, Li L. Modulation of Short-Chain Fatty Acids as Potential Therapy
Method for Type 2 Diabetes Mellitus. *Canadian Journal of Infectious Diseases*
*and Medical Microbiology*. 2021;2021:6632266; doi: 10.1155/2021/6632266.
- 59. May KS, den Hartigh LJ. Modulation of Adipocyte Metabolism by Microbial
Short-Chain Fatty Acids. *Nutrients*. 2021;13(10):3666.
- 60. Chambers ES, Preston T, Frost G, Morrison DJ. Role of Gut Microbiota-
Generated Short-Chain Fatty Acids in Metabolic and Cardiovascular Health.
*Current Nutrition Reports*. 2018;7(4):198-206; doi: 10.1007/s13668-018-
0248-8.
- 61. Gary DW, Charlene C, Eric ZC, Sarah AS, Rachana DS, Kyle B, et al.
Comparative metabolomics in vegans and omnivores reveal constraints on
diet-dependent gut microbiota metabolite production. *Gut*. 2016;65(1):63; doi:
10.1136/gutjnl-2014-308209.
- 62. Malinowska AM. Easy Diet Screener: A quick and easy tool for determining
dietary patterns associated with lipid profile and body adiposity. *Journal of*
*Human Nutrition and Dietetics*. 2022;35(3):590-604; doi:
<https://doi.org/10.1111/jhn.12973>.
- 63. Bai X, Li F, Li F, Guo L. Different dietary sources of selenium alter meat
quality, shelf life, selenium deposition, and antioxidant status in Hu lambs.

- Meat Science. 2022;194:108961; doi:
<https://doi.org/10.1016/j.meatsci.2022.108961>.
- 64. Chen S, Zhou Y, Chen Y, Gu J. fastp: an ultra-fast all-in-one FASTQ
preprocessor. Bioinformatics. 2018;34(17):i884-i90; doi:
10.1093/bioinformatics/bty560.
- 65. Li H, Durbin R. Fast and accurate short read alignment with Burrows-Wheeler
transform. Bioinformatics. 2009;25(14):1754-60; doi:
10.1093/bioinformatics/btp324.
- 66. Li D, Luo R, Liu C-M, Leung C-M, Ting H-F, Sadakane K, et al. MEGAHIT
v1.0: A fast and scalable metagenome assembler driven by advanced
methodologies and community practices. Methods. 2016;102:3-11; doi:
<https://doi.org/10.1016/j.ymeth.2016.02.020>.
- 67. Noguchi H, Park J, Takagi T. MetaGene: prokaryotic gene finding from
environmental genome shotgun sequences. Nucleic Acids Res.
2006;34(19):5623-30; doi: 10.1093/nar/gkl723.
- 68. Fu L, Niu B, Zhu Z, Wu S, Li W. CD-HIT: accelerated for clustering the next-
generation sequencing data. Bioinformatics. 2012;28(23):3150-2; doi:
10.1093/bioinformatics/bts565.
- 69. Li R, Li Y, Kristiansen K, Wang J. SOAP: short oligonucleotide alignment
program. Bioinformatics. 2008;24(5):713-4; doi:
10.1093/bioinformatics/btn025.
- 70. Nissen JN, Johansen J, Allesøe RL, Sønderby CK, Armenteros JJA, Grønbech

CH, et al. Improved metagenome binning and assembly using deep variational
autoencoders. *Nature Biotechnology*. 2021;39(5):555-60; doi:
10.1038/s41587-020-00777-4.

71. Olm MR, Brown CT, Brooks B, Banfield JF. dRep: a tool for fast and accurate
genomic comparisons that enables improved genome recovery from
metagenomes through de-replication. *The ISME Journal*. 2017;11(12):2864-8;
doi: 10.1038/ismej.2017.126.

72. Parks DH, Imelfort M, Skennerton CT, Hugenholtz P, Tyson GW. CheckM:
assessing the quality of microbial genomes recovered from isolates, single
cells, and metagenomes. *Genome Res*. 2015;25(7):1043-55; doi:
10.1101/gr.186072.114.

73. Chaumeil P-A, Mussig AJ, Hugenholtz P, Parks DH. GTDB-Tk: a toolkit to
classify genomes with the Genome Taxonomy Database. *Bioinformatics*.
2019;36(6):1925-7; doi: 10.1093/bioinformatics/btz848.

74. Buchfink B, Xie C, Huson DH. Fast and sensitive protein alignment using
DIAMOND. *Nature Methods*. 2015;12(1):59-60; doi: 10.1038/nmeth.3176.

75. Kanehisa M, Goto S. KEGG: kyoto encyclopedia of genes and genomes.
*Nucleic Acids Res*. 2000;28(1):27-30; doi: 10.1093/nar/28.1.27.

76. Huerta-Cepas J, Szklarczyk D, Forslund K, Cook H, Heller D, Walter MC, et
al. eggNOG 4.5: a hierarchical orthology framework with improved functional
annotations for eukaryotic, prokaryotic and viral sequences. *Nucleic Acids*
*Res*. 2016;44(D1):D286-93; doi: 10.1093/nar/gkv1248.

- 77. Cantarel BL, Coutinho PM, Rancurel C, Bernard T, Lombard V, Henrissat B.
The Carbohydrate-Active EnZymes database (CAZy): an expert resource for
Glycogenomics. *Nucleic Acids Research*. 2008;37(suppl_1):D233-D8; doi:
10.1093/nar/gkn663.
- 78. Jia B, Raphenya AR, Alcock B, Waglehner N, Guo P, Tsang KK, et al. CARD
2017: expansion and model-centric curation of the comprehensive antibiotic
resistance database. *Nucleic Acids Research*. 2016;45(D1):D566-D73; doi:
10.1093/nar/gkw1004.
- 79. Parks DH, Tyson GW, Hugenholtz P, Beiko RG. STAMP: statistical analysis
of taxonomic and functional profiles. *Bioinformatics*. 2014;30(21):3123-4;
doi: 10.1093/bioinformatics/btu494.
- 80. Bastian M, Heymann S, Jacomy M. Gephi: An Open Source Software for
Exploring and Manipulating Networks. *Proceedings of the International AAAI*
*Conference on Web and Social Media*. 2009;3(1):361-2; doi:
10.1609/icwsm.v3i1.13937.

**Table 1.** Nutrients content of crucian, loach, cat food in this research

Item	Crucian	Loach	Cat food
Nutrients contents, % of DM			
DM (Dry matter)	27.25	27.34	91.78
Ash	11.90	9.84	9.21
EE (Ether extract)	16.63	24.39	18.26
CP (Crude protein)	55.46	59.16	19.01
GE (Gross energy), MJ/kg of	27.13	31.48	25.43

**Table 2.** Diet and nutrients intake of otters fed LSCF (raw loaches supplemented with
 commercial cat food) diet in group A or RC (raw crucian) diet in group B.

Item	Group	
	A (LSCF diet)	B (RC diet)
Diet intake, g/d		
Crucian		500
Loach (84%)	504	
Cat food (16%)	96	
Nutrient intake, g/d		
DM (Dry matter)	225.90	136.25
EE (Ether extract)	49.70	22.66
CP (Crude protein)	98.27	75.56
GE (Gross energy), MJ/d	6.58	3.70

**Figure 1.** The distributions of non-redundant unique genes in samples of two groups.

(A) Flower plot of Venn analysis of the distributions of unique genes in 16 samples.

(B) The correlation analysis between samples based on the distribution of unique
genes.

(C) Classification of unique genes into at kingdom level.

**Figure 2.** The compositions and differences of fecal microbial communities of otters
 from two groups in genus level.

(A) The top 20 gut bacteria average relative abundances at genus level in group A and
 group B.

(B) The top 20 gut bacteria relative abundances at genus level in each sample.

(C) The LDA score (log 10) of differential bacteria at genus level (LDA score > 3.5
 and KW test P -value < 0.001).

(D) The α diversity analysis at genus level including Shannon index, Simpson index,
 Richness index, Chao1 index, Pielou index.

(E) Principal-coordinate analysis (PCoA) plot of bacteria community at genus level
 based on Bray–Curtis distance (n=8), $*P < 0.05$, $**P < 0.01$ by independent-samples T
 test.

(F) Flower plot of Venn analysis of bacteria genera number in 16 samples.

 **Figure 3.** The compositions and structures of fecal microbial communities of otters
 from two groups in species level.

(A) The top 20 gut bacteria average relative abundances at species level in group A
 and group B.

(B) The top 20 gut bacteria relative abundances at species level in each sample.

(C) The LDA score (log 10) of differential bacteria at species level (LDA score > 3.5
 and KW test P -value < 0.001).

(D) The α diversity analysis at species level including Shannon index, Simpson,
 Richness index, Chao1 index, Pielou index.

(E) Principal-coordinate analysis (PCoA) plot of bacteria community at species level
 based on Bray–Curtis distance (n=8), * P <0.05, ** P <0.01 by independent-samples T

test.

(F) Flower plot of Venn analysis of bacteria species number in 16 samples.

(G) Co-occurrence network analysis of bacteria community at genus level in group A
and group B where nodes colored according to modules (1-9). Each node represents
one bacteria genus, each edges represents a strong and significant positive or negative
correlation selected based on the threshold of a Spearman rank correlation coefficient
of $P.adjust < 0.05$ and $| r | \geq 0.8$ between two nodes. The size of each node is
proportional to the degree of the bacteria taxa. The thickness of edges is proportional
to the value of the Spearman correlation coefficient.

(H) Clustering heatmap showing the relative abundances of the top 20 bacteria species
taxa in each sample standardized using Z-score method.

**Figure 4.** The classifications and compositions of MAGs in two groups and the
 analysis of differential MAGs.

(A) Summarization of the number of MAGs annotated into different taxonomic levels.
 (B-D) The number of MAGs annotated in each phylum, genus and species taxon. The
 MAGs could not be annotated were not shown.

(E) The relative abundances of top 20 MAGs in different samples. The top 20
 abundant MAGs were annotated in 4 taxa marked by four different colors in the

legend.

(F) The top 20 MAGs average relative abundances in group A and group B.

(G) Principal-coordinate analysis (PCoA) plot of MAGs within two groups based on
Bray–Curtis distance (n=8), * $P < 0.05$, ** $P < 0.01$ by independent-samples T test.

(H) Co-occurrence network analysis of bacteria community at MAG level where
nodes colored according to modules (1-9). Each node represents one MAGs, each
edges represents a strong and significant positive or negative correlation selected
based on the threshold of a Spearman rank correlation coefficient of $P_{\text{adjust}} < 0.05$
and $|r| \geq 0.6$ between two nodes. The size of each node is proportional to the degree
of the MAGs. The thickness of edges is proportional to the value of the Spearman
correlation coefficient.

(I) Distribution of completeness and contamination of all obtained MAGs.

(J) Extended error bar plot of STAMP analysis presenting the significantly differential
MAGs in two groups.

**Figure 5.** The abundance and the differential analysis of KEGG and eggNOG
 annotation.

- (A) KEGG classifications for the fecal metagenome.
- (B) Principal-coordinate analysis (PCoA) plot of KEGG pathways based on Bray-
 Curtis distance (n=8), * $P < 0.05$, ** $P < 0.01$ by independent-samples T test.
- (C) The compositions and abundances of top 20 KEGG pathways at third level in two
 groups.
- (D) eggNOG classifications for the fecal metagenome.
- (E) Principal-coordinate analysis (PCoA) plot of eggNOG categories based on Bray-
 Curtis distance (n=8), * $P < 0.05$, ** $P < 0.01$ by independent-samples T test.
- (F) The compositions and abundances of top 20 eggNOG functional categories in two
 groups.

(G) Extended error bar plot of STAMP analysis presenting differential KEGG
pathways in two groups.

(H) Extended error bar plot of STAMP analysis presenting differential eggNOG
functional categories in two groups.

**Figure 6.** The abundance and the differential analysis of CAZymes and resistance
 genes in two groups.

(A) CAZymes (Carbohydrate-active enzymes) classifications for the fecal
 metagenome.

(B) Compositions and relative abundances of top 20 Carbohydrate-active enzymes
 annotated by CAZy database.

(C) Principal-coordinate analysis (PCoA) plot of CAZyme families based on Bray-
 Curtis distance (n=8).

(D)The relative abundance of GH13 CAZymes in two groups, * $P < 0.05$, ** $P < 0.01$ by
 Mann-Whitney U test.

(E) Clustering heatmap representing the relative abundances of the top 30 resistance
 genes in two groups standardized using Z-score method.

(F) Extended error bar plot of STAMP analysis presenting differential CAZymes in
 two groups

(G) Extended error bar plot of STAMP analysis presenting differential resistance
 genes in two groups.

**Figure 7.** The VFAs concentrations and the correlation analysis between the VFAs
 concentrations and the differential bacteria.

(A) The concentrations of Acetate, Propionate, Butyrate, Isobutyrate, Isovalerate.
 * $P < 0.05$ by independent-samples T test followed by Mann-Whitney U test.

(B) The abundance of *g_Butirivibrio* in two groups, * $P < 0.05$, ** $P < 0.01$ by Mann-
 Whitney U test.

(C) The relative abundance of *g_Roseburia* in two groups, * $P < 0.05$, ** $P < 0.01$ by
 Mann-Whitney U test.

(D) Spearman correlation analysis between the VFAs concentrations and the
 differential bacteria taxa relative abundances at species level. Red denotes a positive
 correlation and blue denotes a negative correlation. The intensity of the color is
 proportional to the strength of Pearson correlation. * $P < 0.05$, ** $P < 0.01$.

Response to Reviewers' Comments

Dear Editor and reviewers:

We sincerely thank the editor and all reviewers for their valuable feedback that we have used to improve the quality of our manuscript. We have addressed the questions point-by-point in the original order. The reviewer comments are laid out below in black text and specific concerns have been numbered. Our responses and changes/additions to the manuscript are given in blue text.

Response to Reviewer # 1

General comments:

In this manuscript, the authors detected the microbiota in feces from otters with metagenomic sequencing and analyzed the microbial functions. This article focuses on the impact of artificial feeding strategies for captive otters on their gut microbiota and also discovers the enrichment of pathogenic microorganisms and resistance genes in otter gut microbiota, thereby pointing out potential problems in the current captive otter feeding process. This work provides new insight and opinion into the gut microbiota of captive otters under different feeding strategies and is very worthy of study and discussion. The manuscript is well-organized and clearly stated. However, there are still some shortcomings in this article.

Response: We thank the reviewer for positive remarks on our manuscript. We tried our best to improve our manuscript according to your constructive comments.

Specific comments:

- 1) The section on "Importance" at the beginning of the manuscript does not provide an accurate and sufficient summary of the main points studied and discussed in the article. There are certain linguistic errors and grammatical mistakes in this part, such as the mistakes in tense and sentence subject in the first sentence, which are lines 42 to 44 of the manuscript.

Response: Thank you for raising the two issues. We have revised the issues you mentioned and have rewritten some of the "Importance" in the manuscript to focus on the effects of

feeding strategies and cat food to otters' gut microbiota and the potential risk of pathogen transmission. The revised "Importance" is as follows:

Captive otters fed with different diet possessed distinct gut microbial communities and functions, with the enrichment of several pathogens and multiple resistance genes in their gut microbiota. The current artificial feeding strategies had the possibility to accelerate the colonization and proliferation of various pathogenic bacteria in the intestines of otters and the spread of resistance genes, increasing risk of diseases. In addition, supplementing commercial cat food had benefits for otters' intestinal fermentation and metabolism of gut microbiota.

2) The content from lines 52 to 56 of the article appears somewhat redundant and unnecessary for the first paragraph of the introduction and can be directly deleted or streamlined. The other part of the "Introduction" section of the article regarding the research background can be appropriately modified and refined to ensure a more accurate and direct presentation of your research objectives.

Response: Thank you for this comment. As you said, we have introduced too many unnecessary research backgrounds in the introduction part, which makes the content not concise enough. We have deleted and modified the content of the introduction of the manuscript according to your suggestions to highlight our research focus. For example, we deleted the first paragraph of the introduction about otters, and re-combined and adjusted the contents of the first three paragraphs.

3) In the discussion on the enrichment of resistance genes (lines 571-573), it was mentioned that the raw loaches supplemented with commercial cat food (LSCF) diet are high in protein and fat. However, I believe this is only compared to another diet, otherwise, a clear definition of high protein or a high-fat diet should be established. Therefore, the discussion here should be more rigorous in its wording.

Response: Thank you for pointing out this issue. The expression we used here is not rigorous enough indeed. Actually, as you said, we are here to express that the LSCF diet has a higher

protein and fat content than the other diet, the RC diet, from the results of the measurements of nutrient contents. It is not that this diet is defined as ' high protein and high fat diet ' because of a certain standard. We have corrected this imprecise expression; the modified content is as follows:

The enrichment of nutrient metabolic pathways in group A may come down to the higher fat, protein and energy intake under the LSCF diet compared to group B.

4) In the description of microbial composition, functional composition, and differences in the "Results" section, unnecessary detailed descriptions of the results can be appropriately reduced, while emphasizing and presenting the differences between the two groups and the connections between the results in each section during the discussion.

Response: Thank you for your helpful suggestion. We have re-examined the results section of the manuscript, paying particular attention to some of the redundant descriptions that you mentioned that may make the article less concise and cohesive. Then, we have deleted and refined these contents to make the presentation of the results more focused on the differences between the two groups.

5) In the discussion section of the article, the author believes that the enrichment of various pathogenic bacteria in feces may come from the food of otters. That may not be rigorous, so the authors need to dig deeper.

Response: We appreciate your comment. Indeed, as you said, our expression here is not rigorous and inadequate. We also mentioned in the introduction of this manuscript, gut microbiota could be affected by many factors like food and environment, food is only one of the more critical factors. In order to make the discussion here more rigorous and adequate, we have revised it based to your suggestions. We have added a more sufficient and deeper discussion on the source of opportunistic pathogens in feces from line537 to 542. The added content is as follows:

However, one thing we cannot ignore is that although we have considered food or feeding strategy as the main risk factor for the transmission and enrichment of pathogen and resistance genes in the above discussion, there are still other factors that may produce similar results, such as living environment and artificial water bodies, as well as possible human transmission due to the fact that otters were raised in zoos, having the chance to contact with

people.

- 6) In line 138 of the article, in the Materials and Methods section, there are two font settings for the letter 's' that need to be adjusted.

Response: Thanks for your careful reading. We have adjusted the font of the letter “s” to “Times New Roman” font.

- 7) The citation format of some references in the article is not very standardized, such as paying attention to the spelling of journal names.

Response: Thanks a lot. We have adjusted the citation format according to the ASM Journals’ style and checked and revised the spelling of journal names.

- 8) The layout between the sub-images in the combination figures of results in the article is somewhat dense and not very neat. It is possible to consider splitting or reassembling the images appropriately, such as Figure 5.

Response: Thank you for catching this. This figure has been recombined by adjusting the layout.

- 9) In Figure 7A, it is recommended not to use mmol/L to represent the content of SCFAs, but to use mmol/g instead. This is because feces are solid and using mmol/L is not appropriate, otherwise, the dilution ratio for extracting SCFAs from feces needs to be clarified.

Response: Thank you for raising this important point. We are sorry for this mistake. We have revised the content unit of SCFAs in Figure 7A to $\mu\text{mol/g}$ according to your suggestion. We have also added more specific descriptions in the Materials and Methods section regarding the determination of SCFAs, clarifying the background details of the conversion between two content units. Currently, Figure 7 has been renamed as Figure 5. Due to considerations of conciseness and data necessity, we have removed the Figures 1 and 2 in initial submission from the manuscript and included them in the supplementary materials. The added statement of determination method of VFAs is as follows:

All samples were thawed at 4 °C, then diluted and mixed with distilled water (1 g stool sample diluted with 1 mL distilled water) for 4 h soaking at 4 °C.

Response to Reviewer # 2

1) On line 82, the presentation's logic appears inconsistent with the preceding content. It is assumed that the authors aim to use the example of the reversal effect of commercial dry food on the gut microbiota composition in dogs fed a meat-based diet to emphasize the significant impact of commercial dry food on the gut microbiota composition.

Response: Thank you for pointing out this issue. As you mentioned, the logic of this statement is indeed contradictory to the preceding content. Following your instruction, we have revised the expression of this sentence to be consistent with context logically to emphasize the significant impact of commercial dry food on the gut microbiota composition. The reference cited here mainly explored the changes on the fecal microbiota in healthy adult dogs with the diet change from dry commercial food to high minced beef, to emphasize the changes in fecal microbiota caused by the switch between two diets and the reshaping effect of commercial food on intestinal microbiota. Although there are some similarities between this research and our work, the focuses and contents of the two research are not same actually. We didn't notice the difference between two research in our submission, which caused this problem. The revised statement is shown below for your convenient reading:

A previous study on canines indicated commercial dry food and meat-based diet seemed to induce different fecal microbiota composition and the commercial cat food can reverse changes in fecal microbiota caused by short-term meat-based diet.

2) In Figure 5, parts of B and E are missing confidence intervals for group B. The confidence intervals for group B are missing. To ensure the consistency of the figure, confidence intervals should be added for group B or deleted for the corresponding part of group A.

Response: Thank you for your suggestion. We have examined the inconsistency you mentioned in Figure 5 based on your suggestion. We found that in fact, in the original figure, we added elliptical confidence intervals to both group A and group B. However, due to the concentrated distribution of sample points in group A, the positions of the confidence ellipse and points basically coincide and the range is small, making them appear less obvious. When using R for plotting, we adjusted the parameter values of the confidence interval (the “level” values in the figure below) from 0.9 to 0.99, and found that it did not have a significant impact on the plotting of the confidence ellipse for Group A. Therefore, we still used the

confidence level of 0.9 for plotting. In addition, we have rearranged and laid out this figure based on the suggestion of another reviewer and deleted the boxplot of figure B and E (now the B and D) convenient for the layout adjustments. Additionally, Figure 5 has been renamed as Figure 3. Due to considerations of conciseness and data necessity, we have removed the Figures 1 and 2 in initial submission from the present manuscript and included them in the supplementary materials. The R code for plotting these two PCoA figures is shown below:

```

color=c("#6596AA", "#867FB3")
p6<-ggplot(data=df, aes(x=V1, y=V2))+
  theme_bw()+
  geom_point(aes(color = group), shape=19, size=2.5)+
  theme(panel.grid = element_blank()+
  geom_vline(xintercept = 0, lty="dashed", linewidth=0.5, color = 'grey50')+
  geom_hline(yintercept = 0, lty="dashed", linewidth=0.5, color = 'grey50')+
  geom_text(aes(x=0.45, y=-0.23), label=paste("R = ", round(df_anosim1$sstatistic, 3)), color="black", size=3)+
  geom_text(aes(x=0.45, y=-0.25), label=paste("P = ", round(df_anosim1$s$signif, 3)), color="black", size=3)+
  labs(x=paste0("PCo1 ", "(", pc[1], "%", ")"),
  y=paste0("PCo2 ", "(", pc[2], "%", ")")+
  scale_color_manual(values = color) +
  scale_fill_manual(values = c("#71ABB6", "#7876B1"))+
  stat_ellipse(data=df, geom = "polygon", level=0.9, linetype = 2, linewidth=0.5, aes(fill=group), alpha=0.2, show.legend = T)+
  theme(axis.title.x=element_text(size=12),
  axis.title.y=element_text(size=12, angle=90),
  axis.text.y=element_text(size=10),
  axis.text.x=element_text(size=10),
  panel.grid=element_blank())+ylim(c(-0.25, 0.25))
p6

```

3) I note that the use of 'Group B' and 'LSCF diet group' in the article is confusing, so please standardize the presentation.

Response: Thank you for spotting this issue. We have uniformly modified all expressions of "LSCF diet group" and "RC diet group" to "group A" and "group B" respectively, meantime the expression of "group A/B fed LSCF/RC diet" is also used in some cases, to eliminate confusion caused by inconsistent names and expressions, while also being consistent with the names used in the figures in this article.

4) Nutritional measurements were performed in the article, and I suggest that the authors provide a systematic and detailed description of the nutritional measurements in the 'Results' section.

Response: Thanks for your suggestion. The description of the result of nutritional measurements about Table 1 and Table 2 has been added in the start of "Results" section. Detailed is as follows for your convenient reading:

The major nutrients contents including dry matter (DM), crude protein (CP), ether extract (EE) also called crude fat (CF), ash content and the gross energy (GE) were measured for identifying and comparing the nutrition values of crucian, loach and cat food. According to the results, the loach has the highest EE, CP and GE content, and the crucian has the highest ash content, while the contents of CP and DM in loach and crucian are more similar

compared to the cat food. The DM content of 91.78% in cat food is the highest clearly due to the fact that it is a commercial dry pet food with little moisture. Then, we calculated the daily intake of various nutrients and gross energy for two groups of otters under two different diets based on the nutritional composition of three food sources as Table 2 shown. Obviously, we could see that the daily intakes of various nutrients (DM, EE, CP and GE) by group A otters are all higher than that of group B otters.

Response to Reviewer # 3

General comments:

The authors explored the influences of two different feeding strategies on the gut microbial community and functions of captive otters, thereby giving some advice on the feeding and management of captive otters. The article is well-written, original, and sound. However, some minor modifications and clarifications are necessary to avoid misunderstandings.

Response: We are grateful for your positive comments. We have revised and improved our manuscript according to your helpful comments.

Specific comments:

1) Lines 123-124: In the “Animals, diets and experimental design” of the part of “MATERIALS AND METHODS”, the description of feed amount is 600 g per day in group B, however, there is an inconsistent data, 500 g per day, presented in Table2. Therefore, the authors need to determine which of these two data is correct and make an accurate modification.

Response: We apologize for this mistake. We checked our original data records, confirming that the feeding amount of 500 g/d is the correct value for otters on the LSCF diet in Group B. Therefore, the description of feeding amount of group B otters is incorrect. We have corrected the description of “600 g per day in group B”to “500 g per day in group B” in MATERIALS AND METHODS.

2) Lines 257-258, in the result of “Taxonomic characteristics of gut microbial communities”, I could know that the proportion accounted by bacteria in group B is not as high as that in

group A, only 73.22%, which means other kingdom taxa like Eukaryote and Archaea may also play a role in the gut microbiota. However, there is no result or discussion of these taxa for explaining their composition and possible role. Therefore, I think it is necessary to provide a brief analysis as supplementary materials on the composition and functions of other microorganisms and add a short discussion.

Response: Thank you for bringing up this important point. We also noticed this phenomenon during data analysis and manuscript writing. We found that the vast majority of species annotated to the eukaryotic superkingdom belong to the animals or Metazoa kingdom, with only a very small portion originating from the fungi kingdom. For these species originating from the animal kingdom, we believe it is highly likely that they were introduced into the intestinal contents through the eating of otters, which has no value to our analysis of gut microbiota. At the same time, due to the extremely low content of microorganisms in the fungi kingdom like that in group A, we did not analyze species in the eukaryotic superkingdom and focused our research on analyzing bacterial communities. In order to avoid unnecessary misunderstandings and potential confusion, we have provided a brief explanation in the results section and submitted the original data related to this article, including annotations on eukaryotic species, as supplementary. The added contents in results are presented below for your convenient reading:

(Lines 261-268 in revised manuscript) Due to the fact that bacteria make up the vast majority of the gut microbiota, we will primarily analyze the bacteria in the intestines of otters. Although almost a quarter of the sequences in group B otters have been annotated to the eukaryotic superkingdom, the vast majority of these sequences are believed to originate from species in the animals kingdom, which may be related to otters consuming animal derived food, while only a very small portion of the fungi kingdom species that may be of concern to us (Detailed data were shown in Table S1). Therefore, we will mainly focus on the bacterial community in the intestines of otters in this article.

3) Line 254: In the second paragraph of “Taxonomic characteristics of gut microbial communities”, the percentage content data of some differential bacteria do not seem accurate because the standard deviation in these data is larger than the average value. Is

this a calculation error or is it caused by the fluctuation of the data itself? Please authors confirm if this is an issue.

Response: Thank you for pointing out this issue. We examined our data of bacteria taxonomic content and reanalyzed these data, we identified that there are no calculated mistakes. We found that this may be due to fluctuations in the relative abundance of some microorganisms between samples, such as, the range of the relative abundances of *Cetobacterium* in the samples of group A is from 0.1818% to 18.89%, and also the relative abundances of *Escherichia_coli* ranges from 1.456% to 35.27%. The large fluctuations in data like this can have an impact on the value of the standard deviation. Therefore, we believe that the fluctuation in the relative abundance of microorganisms in the sample is the reason for the large standard deviation of the data. Additionally, based on the suggestion of another reviewer, we have reduced the unnecessary listing and description of the data, including the listing and display of microorganisms and their relative abundance. Therefore, some of the data that you mentioned that originally had similar issues are no longer listed in the latest submitted manuscript.

4) Lines 442-446: The discussion of α diversity I think is not fully correct and enough. There is a little contradiction in the result between Simpson and Richness index, but the current discussion cannot explain this contradiction, so please authors add some more effective and sufficient discussion about this result.

Response: Thank you for your comment. Our explanation in the initial manuscript was indeed insufficient or even incorrect. Based on your suggestion, we have revised and supplemented the explanation of alpha diversity, especially the conflict between Richness and Simpson index. After our reanalysis and discussion, we believe that there is not a complete conflict between these two indicators. Richness index simply determines the number of bacterial genera contained in the sample, while Simpson index can reflect both bacteria richness and evenness, especially evenness. Therefore, this means that the higher Simpson index value in Group A is due to its microbial community having better uniformity. The revised statement is as follows:

(Lines 440-447 in revised manuscript) Herein, we compared the gut microbial communities

that existed in two groups of otters, observing apparently different microbial compositions and higher Simpson index at genus and species level in the gut microbiota of group A otters, whereas the Richness index was higher in group B at genus level, indicating the microbial community in group B possesses richer bacteria genus taxa, while the evenness of gut microbiota in group A might be higher considering that the Simpson index can reflect the two dimensions that richness and evenness in a community, especially with a more sensitive response to evenness.

- 5) Figure 7D: the name of this image in the caption is described as Spearman correlation, while the description in the last sentence, line 1025, is written as Pearson correlation. Please clarify the reasons for the inconsistency between the two statements, and also determine whether the analysis used the correct method. In my opinion, Spearman correlation analysis should be used for this analysis.

Response: Thanks for pointing out this mistake. We have corrected this mistake. Actually, we used Spearman correlation in our analysis. The reason for this mistake is that we initially used Pearson correlation in our analysis, but later found that this analysis method was not suitable. Therefore, we used Spearman correlation analysis to process the data. However, after completing the modification, we forgot to modify the description in the caption of the figure and only changed the name of the analysis method in the name of the figure. We apologize for this mistake caused by carelessness. Additionally, Figure 7 has been renamed as Figure 5. Due to considerations of conciseness and data necessity, we have removed the Figures 1 and 2 in initial submission from the manuscript and included them in the supplementary materials.

- 6) Discussion: The listing and introduction of pathogenic bacteria in the discussion can be appropriately reduced, as some of the listings are not very necessary, otherwise it will make the discussion section redundant, located in this manuscript. I suggest swapping the positions of the second to last and third to last paragraphs in the discussion section to ensure continuity and coherence of the discussion.

Response: Thanks for your valuable suggestions. We have removed some unnecessary discussions based on your suggestion, especially the some of the content introducing

microorganisms, and have also swapped the positions of the two paragraphs you mentioned.

7) Abstract: One or two sentences of brief introductions about the research background could be added to explain why you carry out this research.

Response: Thank you for the suggestion. As you mentioned, the background information is needed indeed to benefit the readers to quickly understand our work. We have added one sentence of brief introduction of our research background in the beginning of abstract based on your suggestion. The added statement as follows:

Captive otters raised in zoos are fed different artificial diets, which may shape different gut microbiota.

Thank you again for your comments! We appreciate for Editors' and Reviewer's warm work earnestly, and hope that the correction will meet with approval.

Once again, thank you very much for your comments and suggestions.

Respectfully yours,

Corresponding author: Long Guo (guolong@lzu.edu.cn)

State Key Laboratory of Grassland Agro-ecosystems, College of Pastoral Agriculture Science and Technology, Lanzhou University

Lanzhou, Gansu, China

Re: mSystems00954-24R1 (Different artificial feeding strategies shape the diverse gut microbial communities and functions with the potential risk of pathogen transmission to captive Asian small-clawed otters (*Aonyx cinereus*))

Dear Dr. Long Guo:

Your manuscript has been accepted, and I am forwarding it to the ASM production staff for publication. Your paper will first be checked to make sure all elements meet the technical requirements. ASM staff will contact you if anything needs to be revised before copyediting and production can begin. Otherwise, you will be notified when your proofs are ready to be viewed.

Sincerely,
Suzanne Ishaq
Editor
mSystems

Reviewer #1 (Comments for the Author):

The author has addressed all my comments.

Reviewer #2 (Comments for the Author):

The manuscript has undergone comprehensive and meticulous revisions to address all the feedback provided by the reviewers.